# Stepwise differentiation of follicular helper T cells reveals distinct developmental and functional states

Manuel A. Podestà [1,2,6], Cecilia B. Cavazzoni[1,6], Benjamin L. Hanson[1], Elsa D. Bechu[1], Garyfallia Ralli[1], Rachel L. Clement [1], Hengcheng Zhang[1], Pragya Chandrakar[1], Jeong-Mi Lee[1], Tamara Reyes-Robles[3], Reza Abdi[1], Alos Diallo[4], Debattama R. Sen [5] & Peter T. Sage [1] ✉

Follicular helper T (Tfh) cells are essential for the formation of high affinity antibodies after vaccination or infection. Although the signals responsible for initiating Tfh differentiation from naïve T cells have been studied, the signals controlling sequential developmental stages culminating in optimal effector function are not well understood. Here we use fate mapping strategies for the cytokine IL-21 to uncover sequential developmental stages of Tfh differentiation including a progenitor-like stage, a fully developed effector stage and a post-effector Tfh stage that maintains transcriptional and epigenetic features without IL-21 production. We find that progression through these stages are controlled intrinsically by the transcription factor FoxP1 and extrinsically by follicular regulatory T cells. Through selective deletion of Tfh stages, we show that these cells control antibody dynamics during distinct stages of the germinal center reaction in response to a SARS-CoV-2 vaccine. Together, these studies demonstrate the sequential phases of Tfh development and how they promote humoral immunity.

High affinity effector antibodies are generated in germinal centers (GC), microanatomical structures in lymphoid organs that promote efficient interactions between B cells and T follicular helper (Tfh) cells[1,2]. Tfh cells provide costimulatory signals and cytokines to B cells, promoting class switch recombination, somatic hypermutation, and affinity maturation. These signals also induce GC B cells to differentiate into antibody-secreting long-lived plasma cells or memory B cells. T follicular regulatory (Tfr) cells dampen Tfh-mediated B cell activation, thereby restraining autoreactive B cells, and to a lesser extent foreign antigen-specific B cells, from participating in the GC reaction[3–6].

The complete Tfh cell transcriptional program is thought to only develop after multiple interactions with antigen-presenting cells in secondary lymphoid organs. This process entails differentiation into a pre-Tfh phenotype elicited by dendritic cells, encounter of Tfh with B cells at the T-B border, and full differentiation into a GC Tfh cell only after further interactions with cognate B cells. Despite this paradigm, more recent data suggest that only a small percentage of fully differentiated Tfh cells actually reside within GCs themselves[7]. After interaction with B cells in GCs, Tfh cells may undergo cell death or become dysfunctional to extinguish the GC response[8]. Moreover, some Tfh cells can bypass the B cell zone altogether and enter the circulation as a memory-like population, which can become reactivated in other lymphoid organs to provide a quicker secondary response[9]. Intrinsic factors that control initial differentiation of naïve CD4 T cells to Tfh cells have been identified, including both positive (Bcl6, Ascl2, MAF, Tcf1, Lef1) and negative regulators (Prdm1, Foxo1, Foxp1, Id2, and Klf2)[2,10].

[1]Transplantation Research Center, Renal Division, Brigham and Women's Hospital, Harvard Medical School, Boston, MA, USA. [2]Renal Division, Department of Health Sciences, Università degli Studi di Milano, Milano, Italy. [3]Exploratory Science Center, Merck & Co., Inc., Cambridge, MA 02141, USA. [4]Department of Immunology, Harvard Medical School, Boston, MA, USA. [5]Center for Cancer Research, Massachusetts General Hospital, Harvard Medical School, Charlestown, MA, USA. [6]These authors contributed equally: Manuel A. Podestà, Cecilia B. Cavazzoni. ✉e-mail: psage@bwh.harvard.edu

However, factors that control stepwise development through putative Tfh differentiation transitory states after Tfh commitment have been less studied.

The nature of the inflammatory challenge promoting Tfh differentiation can instruct secretion of proinflammatory cytokines that polarize antibody isotypes to optimize immunity. In some cases, cytokine-polarized Tfh cells positively correlate with B cell responses in disease states, suggesting the unique functionality of these Tfh subsets[6,11–16]. The cytokine IL-21 produced by Tfh cells promotes B cell responses in a paracrine manner, as exemplified by the reduction of antibody-secreting cells, GC persistence, and somatic hypermutation observed in experimental germline deficiency of IL-21 or its receptor[11–15]. Although IL-21 is the prototypical Tfh cytokine, previous studies suggested that only a fraction of Tfh cells produce IL-21, and that these cells replace IL-21 with IL-4 over time[16,17]. Recent data also suggest that GC-resident and non-resident Tfh cells have similar expression of IL-21, at least at the transcriptional level, suggesting that anatomical location does not dictate IL-21 status[7]. Moreover, Tfr cells may suppress IL-21 production in Tfh cells, thereby masking IL-21 competent cells[18]. Therefore, the origins, developmental trajectories, and functionality of Tfh stages remain largely unknown.

Here we used a series of genetic tools to track the developmental stages and analyze the functions of Tfh cells in vivo. We uncovered that Tfh differentiation occurs in distinct sequential developmental stages marked by the production of IL-21 and that these stages are transcriptionally and epigenetically regulated as well as largely independent of anatomic location. Furthermore, we found that commitment to the fully developed Tfh stage is terminal and that these cells can extinguish IL-21 expression but remain epigenetically poised. We also found that Tfr cells extrinsically, and FoxP1 intrinsically, control the balance of stem-like progenitor cells and effector Tfh cells. In vivo deletion of Tfh cells in later developmental stages at distinct time points of the GC response during SARS-CoV-2 Spike-protein vaccination revealed that these cells promote GC formation, prevent GC contraction, and enhance somatic hypermutation. Our results reveal distinct stepwise developmental stages after Tfh commitment as well as the intrinsic and extrinsic mechanisms controlling stage transitions to optimize humoral immunity.

## Results

### Tfh cells previously expressing IL-21 are transcriptionally and phenotypically distinct from Tfh progenitor-like cells

To understand the development of Tfh cells, we utilized a system to track the historical expression of the cytokine IL-21 as an indicator of near-terminal development. Alternative populations of near-terminal development, such as PD-1[hi] or Bcl6[hi] Tfh cells, were not possible to identify in a historical manner due to a lack of genetic tools. We crossed *Il21*[Cre] mice to a *Rosa26*[Lox-STOP-Lox-YFP] fate mapping (FM) strain to allow lineage-tracing of cells that have ever expressed IL-21[19]. We immunized IL21-FM (*Il21*[Cre]*Rosa26*[Lox-STOP-Lox-YFP]) mice with 4-hydroxy-3-nitrophenyl-acetyl-ovalbumin (NP-OVA) and harvested draining lymph nodes (dLNs) and blood 5, 8 or 11 days later. Tfh cells were defined as CD4[+]CD19[-]GITR[-]ICOS[+]CXCR5[+] and then subdivided into YFP-expressing and non-expressing cells (Fig. 1a). Although expression of CXCR5 and ICOS for Tfh gating has limitations due to potential expression after early activation, this was the only available strategy for these studies. We referred to YFP[+] cells within Tfh gated cells as fully developed Tfh (Tfh-Full) cells, and YFP[-] within the Tfh gate as Tfh progenitor (Tfh-Prog) cells, since they are phenotypically a Tfh population but have never expressed IL-21. In the dLN, total Tfh cells peaked at day 8 when Tfh-Full cells comprised ~65% of all Tfh cells. In the blood, however, Tfh-Full cells increased at day 5 after immunization, and their frequency consistently remained ~75% of Tfh cells over time. Therefore, IL-21 fate-mapped Tfh-Full cells increase over time and are found in both the dLN and blood after vaccination. Tfh-Full cells

were also found in the spleen, but the proportion of these cells only changed incrementally over the course of immunization. In contrast to Tfh-Full cells, the frequency of IL-21 fate-mapped T conventional (Tcon21) cells, defined as CD4[+]CD19[-]GITR[-]CXCR5[-]YFP[+], did not increase over time in the dLN, but did increase in the blood (Fig. 1b). To confirm that Tfh-Prog and Tfh-Full both respond to vaccine-specific antigens we adoptively transferred cell trace violet (CTV) labeled CD4[+] T cells (from OT-II[+]*Il21*[Cre]*Rosa26*[Lox-STOP-Lox-YFP] mice) into NP-OVA immunized WT mice and 8 days later assessed CTV in Tfh-Prog and Tfh-Full gated cells. Almost all cells in the Tfh-Prog and Tfh-Full gates showed evidence of proliferation suggesting a response to antigen, although Tfh-Full cells underwent more rounds of division than Tfh-Prog cells (Fig. 1c and Fig. S1a–b). PD-1 expression was highest on the most proliferated OT-II cells regardless of whether they were Tfh-Prog or Tfh-Full (Fig. S1). Moreover, although PD-1[hi] OT-II Tfh cells were enriched for Tfh-Full cells, a substantial population of Tfh-Prog was also contained in this gate, suggesting that while proliferation/activation may be necessary for developmental stage progression, it is not sufficient and other factors may dictate developmental fate decisions (Fig. S1).

To assess the transcriptional identity of Tfh-Full and Tfh-Prog cells in more detail at the peak of the Tfh response, we used bulk RNAseq transcriptional analysis. IL21-FM mice were vaccinated with NP-OVA and dLNs were harvested on day 7. We sorted four populations based on Tfh phenotype and previous IL-21 expression (i.e. conventional CD4[+] T cells (Tcon), Tcon21, Tfh-Prog, and Tfh-Full cells). By principal component analysis (PCA), both previous IL-21 production and the Tfh phenotype contributed to the separation of the four populations (Fig. 1d). To compare the core Tfh signature in Tfh-Prog versus Tfh-Full cells, we performed Gene Set Enrichment Analysis (GSEA) using a LN Tfh signature from a previously published study[20]. Tfh-Full cells showed a stronger enrichment for the core Tfh signature (NES = 1.75, $p < 0.001$) compared to Tfh-Prog cells (NES = 1.28, $p = 0.006$), suggesting that the former are more fully differentiated than Tfh-Prog cells (Fig. 1e). When we compared differentially expressed genes (DEGs) in Full versus Prog cells we found 521 upregulated genes in Tfh-Full cells and 242 upregulated genes in Tfh-Prog cells (Fig. 1f). Upregulated genes in Tfh-Full cells included *Il21*, *Maf*, *Cd44*, *Pdcd1* and *Icos*. When we assessed genes positively and negatively associated with Tfh cells we found that Tfh-Full cells expressed higher levels of selected positive regulators of Tfh cells (Fig. 1g). These included genes such as *Sh2d1a*/SAP, *Bcl6*, *Tcf7*, and *Tox1/2*, all of which have been associated with the promotion of Tfh cells. In contrast, two genes that have important functions in promoting Tfh development, *Pcyt2*, and *Lef1*, were higher in Tfh-Prog cells[21,22]. Some genes negatively associated with Tfh cell development, such as *Bach2* and *Satb1*, were higher in Tfh-Prog compared to Tfh-Full cells[23,24]. In contrast, *Hif1a* was more highly expressed in Tfh-Full compared to Tfh-Prog cells[25]. Tfh-Prog cells expressed higher levels of naïve-like transcripts and proteins such as *Sell*/CD62L, but still maintain substantial ICOS expression (Fig. 1g, h). Tfh-Full cells expressed higher levels of ICOS and CD44 compared to Tfh-Prog cells at the protein level at all time points analyzed. To assess the anatomical location of Tfh-Full cells we performed immunofluorescence analysis on dLNs. YFP[+] cells in the B cell follicle were found inside and outside of GCs, suggesting that Tfh-Full cells can be present throughout the B cell follicle (Fig. 1i). Together, these data suggest that IL-21 fate-mapped Tfh-Full cells, which are found in lymph nodes and blood, are phenotypically and transcriptionally distinct from Tfh-Prog cells, suggesting these two cell states mark distinct stages of Tfh development.

### Tfr cells regulate the Tfh-Prog to Tfh-Full transition to control germinal center responses

Tfr cells can suppress Tfh cells after differentiation through regulation of IL-21 production[18]. To identify possible ways in which the immune

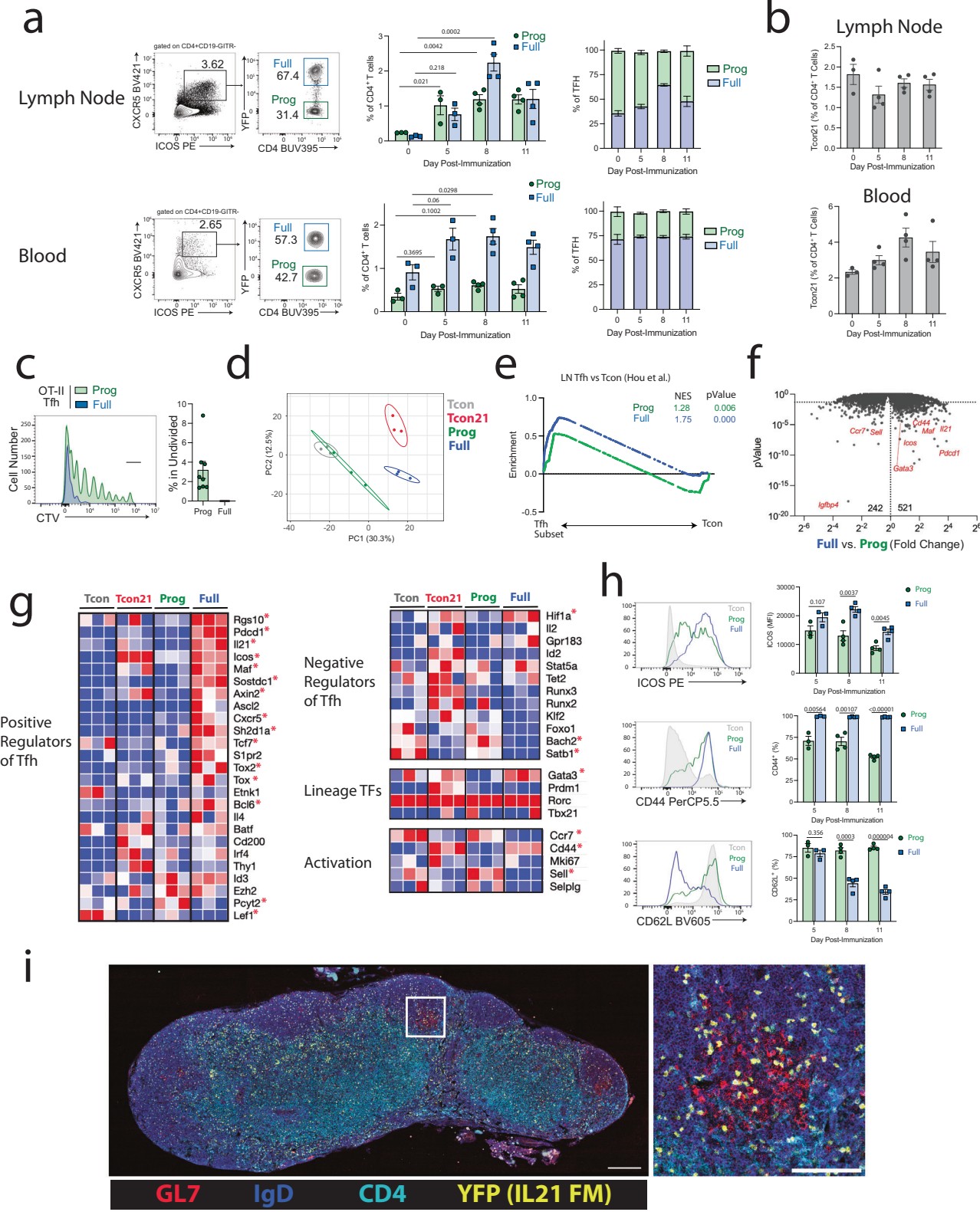

system can regulate Tfh-Full differentiation, we assessed the effect of Tfr cells on the Tfh-Prog to Tfh-Full developmental transition. We reanalyzed a previously published bulk RNA-seq dataset in which total Tfh cells were suppressed by Tfr cells in vitro[18]. In this dataset, "activated" and Tfr-"suppressed" Tfh cells segregated by PCA analysis (Fig. 2a). We generated a gene set that identifies genes more highly expressed in Tfh-Prog versus Tfh-Full cells from bulk RNA-seq (as

shown in Fig. 1d), and then assessed enrichment of this gene set within the "activated" or Tfr-"suppressed" Tfh phenotype. We found that the Tfh-Prog phenotype was substantially enriched in the Tfr-"suppressed" Tfh condition, suggesting transcriptional similarities between these two states (Fig. 2b). Overall, we found 192 genes that were differentially expressed both in Tfr-"suppressed" versus "activated" Tfh cells and Tfh-Prog versus Tfh-Full cells (Fig. 2c). Some of these genes include

**Fig. 1 | Tfh-full cells are phenotypically distinct from Tfh-Prog cells.**
**a** *Il21*[Cre]*Rosa26*[Lox-STOP-Lox-YFP] mice were immunized with NP-OVA and tissues collected.
(Left) Gating strategy for Tfh-Prog ("Prog") and Tfh-Full ("Full") cells. (Middle)
Quantification of Tfh-Prog and Tfh-Full cells over time. (Right) Distribution of Tfh-
Prog and Tfh-Full cells of all Tfh cells (*n* = 3 for 0,5d and *n* = 4 for 8,11d). **b** Frequency
of "Tcon21" (CD4 + CXCR5-YFP+) cells in lymph nodes and blood from mice as in
(**a**). **c** Cell trace violet (CTV) labeled CD4 T cells from OT-II*Il21*[Cre]*Rosa26*[Lox-STOP-Lox-YFP]
mice were transferred to WT mice which were NP-OVA immunized and organs
harvested on day 9 (*n* = 8). **d** PCA plot of indicated populations from NP-OVA
immunized *Il21*[Cre]*Rosa26*[Lox-STOP-Lox-YFP] mice using bulk RNAseq transcriptional data.
(*n* = 3 independent experiments shown, each included 10 mice per group). **e** GSEA
enrichment of a "Tfh gene module" (from Hou et al.) in Tfh-Prog vs. Tcon or Tfh-Full

vs. Tcon. NES= normalized enrichment score. *P* values were calculated with
empirical phenotype-based permutation tests. **f** Volcano plot comparing genes in
Tfh-Full versus Tfh-Prog cells from RNAseq data in (**d**)). *P* values were calculated
using EdgeR. **g** Heatmap of genes involved in indicated pathways. Asterisks indicate
genes significantly different (*P* < 0.05) between Tfh-Prog and Tfh-Full cells from (**f**).
**h** Expression of ICOS, CD44 or CD62L in Tfh-Prog and Tfh-Full cells at indicated
times post vaccination. (*n* = 3 for 5d and *n* = 4 for 8,11d). **i** Micrograph of dLN from
NP-OVA immunized *Il21*[Cre]*Rosa26*[Lox-STOP-Lox-YFP] mouse. Scale bars (left, 250 μM; right
100 μM). Micrograph is from one experiment which is representative of 3 inde-
pendent repeats. **a**, **b**, **c**, **h** Data are represented as mean ± s.e.m. **a** One-way ANOVA
with Dunnett's correction. **c**, **h** Unpaired two-tailed student's *t*-test.

---

*Sell*, *Ccr7* and *Foxp1*, the latter of which has been shown to regulate
initial Tfh differentiation from naïve conventional CD4⁺ T cells[26]. Sev-
eral metabolically related genes were also upregulated both in acti-
vated versus suppressed Tfh and Tfh-Full versus Tfh-Prog cells. These
data suggest similarities of Tfh-Prog and Tfr-suppressed Tfh cells.

To assess whether Tfr cells can regulate Tfh-Full cells, we next
sought to assess the effect of Tfr cell deletion on Tfh-Full cells in vivo.
To do this we used a Tfr-DTR mouse model (*Foxp3*[Cre]*Cxcr5*[Lox-STOP-Lox-DTR]),
which has been published previously[5,6,27]. We crossed Tfr-DTR mice to
an *Il21*[VFP] strain (previously published[28]) to report IL-21 expression,
since we could not incorporate the IL-21 fate mapping allele due to
incompatible genetic strategies. We vaccinated these mice with NP-
OVA, deleted Tfr cells with administration of diphtheria toxin (DT) and
harvested dLNs on day 10 post-immunization (Fig. 2d). Tfr cells were a
substantially smaller population of all CXCR5-expressing CD4⁺ T cells
in Tfr-DTR mice, consistent with deletion of Tfr cells and previous
reports of deletion potency (Fig. 2e)[5,6,27]. We found increases in the
frequencies of both Tfh-Prog (CD4⁺CD19⁻GITR⁻PD1⁺CXCR5⁺VFP⁻) as
well as Tfh-Full (CD4⁺CD19⁻GITR⁻PD1⁺CXCR5⁺VFP⁺) cells as a frequency
of all CD4⁺ T cells, suggesting that Tfr cells control differentiation
stages in Tfh cells (Fig. 2f).

The higher frequency of Tfh-Full cells after Tfr deletion may be
explained by multiple mechanisms. Tfr deletion may lead to an
increase in Tfh-Prog cells that ultimately give rise to Tfh-Full, Tfr may
directly limit the Tfh-Prog to Tfh-Full developmental transition, or
both. To specifically assess whether Tfr cells can alter the Tfh-Prog to
Tfh-Full developmental transition, we used a modified version of a
previously published Tfr suppression assay in which we could syn-
chronize Tfr suppression and control the developmental stage of
input cells[6,29]. We sorted Tfr (CD4⁺ICOS⁺CXCR5⁺FoxP3⁺) and total B
(CD19⁺CD4⁻) cells from *Foxp3*[GFP]*Ptprc*[a] (CD45.1⁺) mice 7 days after
immunization with NP-OVA/CFA (Fig. 2g). We cultured these cells
with Tfh-Prog or Tfh-Full cells from *Il21*[Cre]*Rosa26*[Lox-STOP-Lox-YFP] mice. In
this system, the use of a fate-mapping allele ensures that Tfh-Full
presence in Tfh-Prog cultured conditions is due to de novo differ-
entiation of Tfh-Full cells. After 6 days of culture, we identified the
Tfh input cells as CD45.1[neg] and assessed de novo IL-21 expression
through the YFP allele. We observed a clear population of Tfh-Full
cells that differentiated from Tfh-Prog cells in conditions that did not
contain Tfr cells (Fig. 2h, i). Wells that contained Tfr cells had a sig-
nificant reduction in YFP⁺ Tfh-Full cells both by percentage and total
number, suggesting that Tfr cells inhibit the differentiation of Tfh-
Prog to Tfh-Full cells. We also assessed the roles of Tfr cells in sup-
pressing the expansion of already differentiated Tfh-Full cells using
similar in vitro assays, but with Tfh-Full cells as input. We found
reduced expansion of Tfh-Full in the presence of Tfr compared to
control wells, suggesting Tfr cells can also suppress Tfh-Full expan-
sion (Fig. 2j, k). Taken together, these data indicate that Tfr cells
control multiple developmental stage transitions in Tfh cells, since
they can inhibit the initial differentiation of naïve CD4⁺ to Tfh-Prog,
the differentiation of Tfh-Prog to Tfh-Full cells, and Tfh-Full
expansion.

## The stepwise developmental stages of Tfh differentiation are transcriptionally programmed

To study Tfh-Full developmental stages in more detail we crossed
transgenic *Il21*[cre] mice with a *Rosa26*[Lox-STOP-Lox-tdTomato] allele and the *Il21*[VFP]
knock-in reporter strain to generate the IL21[FM/Rep] (fate mapper/
reporter) strain. In the IL21[FM/Rep] strain, cells that *currently* produce IL-21
are marked with VFP expression and cells that *ever* expressed IL-
21 are marked with tdTomato. When assessing total Tfh cells in the dLN
of these mice after vaccination we found four distinct populations
defined by tdTomato and VFP expression (Fig. 3a). These include Tfh-
Prog cells (progenitor: TdTomato⁻VFP⁻) that have never produced IL-21,
Tfh-Full cells (fully developed: TdTomato⁺VFP⁺) that currently produce
IL-21 and express the fate mapping allele, Tfh-Ex cells (ex IL-21:
TdTomato⁺VFP⁻) that do not currently express IL-21 but have in the
past, and a small population of Tfh-Trans cells (transitory:
TdTomato⁻VFP⁺), which have only recently started to express IL-21 and
are not yet marked by tdTomato (due to a temporal gap from when IL-
21 is expressed and when Cre recombinase can mediate excision of the
transcriptional stop cassette). In the dLN, Tfh-Prog and Tfh-Full cells
were most substantial, with a small, but meaningful, population of
both Tfh-Trans and Tfh-Ex cells present (Fig. 3a). Localization of the
four populations by microscopy revealed that Tfh-Full cells were
enriched in the GC, but that all four populations could be found in the
GC and surrounding follicle demonstrating Tfh developmental stages
are not largely dictated by anatomical location (Fig. 3b). To confirm
that all four populations can differentiate and persist at multiple
timepoints in response to vaccine antigens we adoptively transferred
CD4⁺ T cells from naïve OT-II⁺IL21[FM/Rep] mice into WT immunized mice
and assessed populations over time. The frequency of Tfh-Trans
peaked at day 5 and steadily decreased over time, whereas the fre-
quency of Tfh-Ex increased over time (Fig. 3c). Importantly, all four
populations were found at all timepoints, albeit at different ratios,
suggesting developmental trajectories and not stochastic
development.

To determine if developmental stages were transcriptionally dis-
tinct, we immunized IL21[FM/Rep] mice with NP-OVA, and 9 days later
sorted Tfh-Prog, Tfh-Trans, Tfh-Full, Tfh-Ex and Tcon
(CD4⁺GITR⁻PD1⁻CXCR5⁻) cells, which were marked with cell hashing
antibodies and loaded on a 10X chromium platform to perform single
cell RNAseq analysis. Unbiased clustering based on gene expression
UMI counts of the combined dataset revealed 8 distinct clusters after
dimensionality reduction, consistent with significant heterogeneity in
Tfh cells (Fig. 3d and Fig. S2a–c). Assessment of sample hashtags
demonstrated that Tfh developmental stages were not confined to a
single cluster and in fact, some clusters contained multiple Tfh
developmental stage cells. Tfh-Full and Tfh-Prog were largely non-
overlapping suggesting distinct transcriptional programming. Tfh-
Trans and Tfh-Ex cells were found in multiple clusters with similar
dispersal pattern suggesting some transcriptional similarity. A small
population of Tfh-Ex cells were found in cluster 8 and expressed
FoxP3 suggesting that a very small (~3%) percentage of Tfh-Ex may
upregulate FoxP3 (Fig. S2f–h), a phenomenon that has recently been

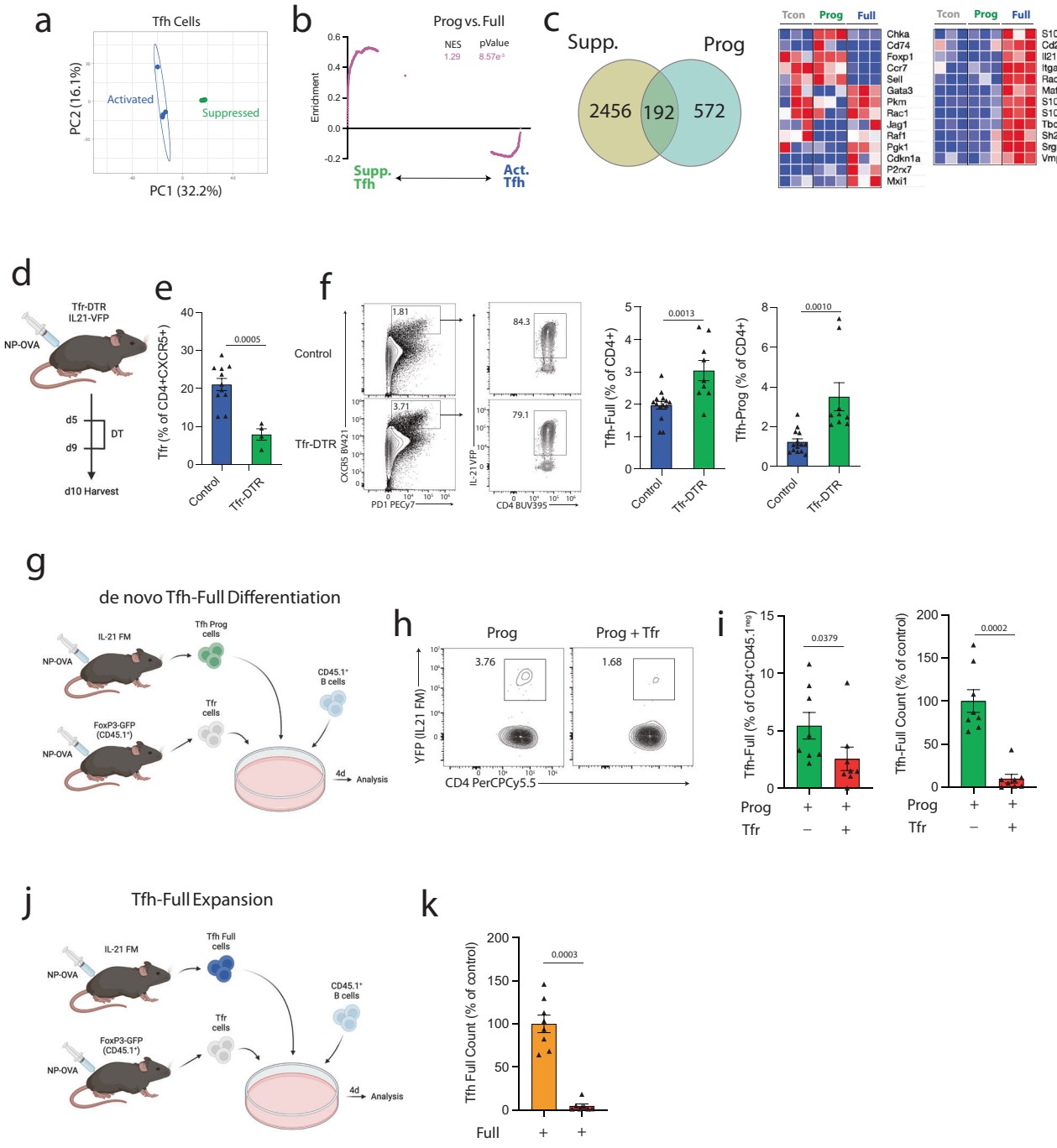

**Fig. 2 | Tfr cells regulate the Tfh-Prog to Tfh-Full transition. a** Principal component analysis (PCA) plot showing activated ("Activated") or Tfr-suppressed ("Suppressed") Tfh cells using RNASeq data from Sage et al.[18]. **b** Gene set enrichment analysis (GSEA) of Activated or Suppressed Tfh cells using a Tfh-Prog vs. Tfh-Full gene set (from RNAseq in Fig. 1). NES= normalized enrichment score. *P* value was calculated using empirical phenotype-based permutation tests (GSEA). **c** (Left) Diagram demonstrating genes differentially expressed (*P* < 0.05 calculated using EdgeR) in both Suppressed versus Activated Tfh cells (Sage et al.[18]) and Tfh-Prog versus Tfh-Full cells. (Right) Heatmap of RNASeq data for a subset of genes from (**c**). **d** IL-21 reporter Control (*Foxp3*^Cre CXCR5^WT *Il21*^VFP) or Tfr-DTR (*Foxp3*^Cre CXCR5^Lox-STOP-Lox-DTR *Il21*^VFP) mice were immunized with NP-OVA and diphtheria toxin (DT) given to delete Tfr cells. DLNs were harvested on day 10. **e** Frequency of Tfr cells from (**d**). (Con *n* = 11, DTR *n* = 4), and is combined data from 2 independent experiments, error bars indicate mean ± s.e.m.). *P* value was calculated using two-tailed unpaired Student's *t*-test. **f** Gating strategy (left) and

quantification (middle) of Tfh-Full cells or Tfh-Prog cells (right) from experiments as in (**e**) (Con *n* = 14, DTR *n* = 9), and is combined data from 3 independent experiments, error bars indicate mean ± s.e.m. *P* value indicates two-tailed unpaired Student's *t*-test. **g** De novo Tfh-Full differentiation. Tfh-Prog (CD4^+CXCR5^+YFP^−) cells were sorted from immunized IL-21 FM (*Il21*^cre *Rosa26*^YFP) mice and cultured with B and Tfr (CD4+CXCR5+FoxP3+) cells from CD45.1+ *Foxp3*^GFP mice along with anti-IgM and anti-CD3/28 beads. **h** Gating strategy for Tfh-Full cells as in (**g**). Plots are pregated on CD45.1^−CD4^+IA^−. **i** Frequency (left) or count (right) of Tfh-Full cells. (*n* = 8, and is combined data from 2 independent experiments, error bars indicate mean ± s.e.m. *P* values were calculated using unpaired Mann–Whitney test). **j** Schematic to assess Tfh-Full expansion. Tfh-Full (CD4^+CXCR5^+YFP^+) cells were sorted from IL-21 FM (*Il21*^cre *Rosa26*^YFP) mice and cultured. **k** Tfh-Full count (normalized to control mean). (*n* = 8 replicates per group, and is combined data from 2 independent experiments, error bars indicate mean ± s.e.m.). *P* values were calculated using unpaired Mann–Whitney test.

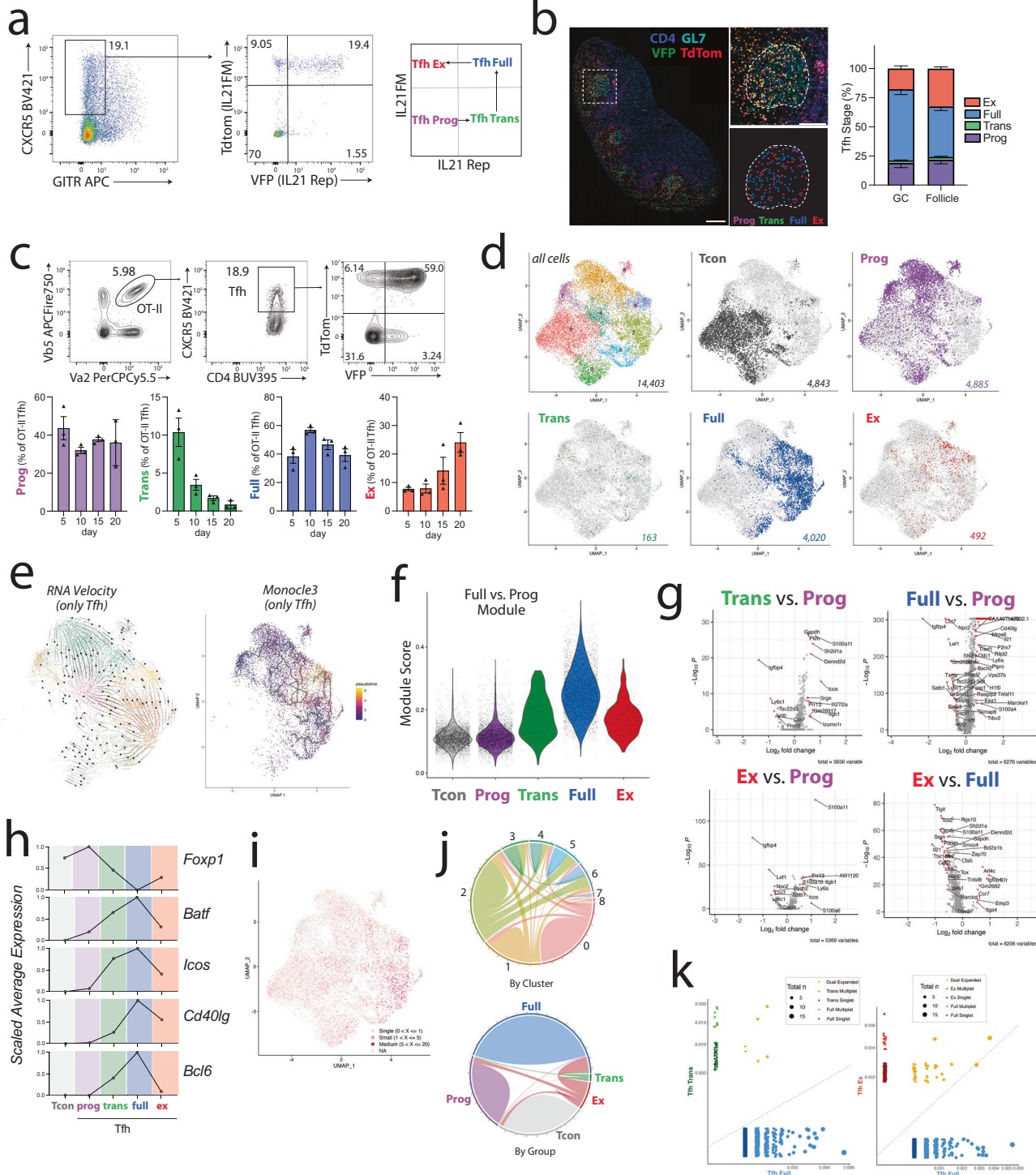

**Fig. 3 | Stepwise differentiation of effector Tfh cells is driven by transcriptional programming. a** Tg(*Il21*^cre^)*Rosa26*^Lox-STOP-Lox-TdTomato^*Il21*^VFP^ ("IL-21^FM/Rep^") mice were immunized with NP-OVA and dLN harvested on day 9. Gating strategy for the four developmental stages of Tfh cells: Progenitor ("Prog"), Transitory ("Trans"), Fully differentiated ("Full"), Ex IL-21 expressors ("Ex"). Plots are pregated on CD4⁺CD19⁻ cells. **b** LNs from experiments as in (**a**) were assessed by microscopy. Representative LN with individual GC is shown. Scale bars = 250 μM (left) and 100 μM (right). Distribution of individual Tfh developmental stages among total Tfh in GC and non-GC B cell follicle are shown (*n* = 6, GC; *n* = 8, Follicle). Data are represented as mean ± s.e.m. **c** CD4⁺ T cells from OTII⁺IL-21^FM/Rep^ mice were transferred to WT mice which were immunized. dLNs were harvested on indicated days. *n* = 3 per group. **d** Uniform Manifold Approximation and Projection (UMAP) of scRNAseq data derived from indicated populations (gated as in (**a**)) from IL-21^FM/Rep^ mice 9 days

after immunization with NP-OVA. All cells and indicated individual groups are shown. Number indicates cells in group. **e** RNA velocity (left) and Monocle pseudo-time analysis (right) including only Tfh developmental stages. **f** Gene module score for a Tfh "Full vs. Prog gene module" derived from bulk RNAseq data in Fig. 1f. **g** Differential gene expression (DESeq2 model implemented in the Seurat FindAll-Markers() function with a log(fold-change) threshold of 0.1) between indicated groups. Ribosomal genes are omitted from plots. **h** Scaled average expression of selected genes across Tfh developmental stages (obtained with the Seurat Aver-ageExpression function) **i** Extent of clonal expansion in dataset. **j** Circos plot showing clonal overlap in cells by cluster or by stage (only expanded clones included). **k** Scatter plots with clonal overlap between indicated Tfh subsets.

demonstrated to occur in Tfh cells during GC contraction[8]. RNA velocity analysis demonstrated predicted movement within clusters including focusing of cells toward a portion of Cluster 2 which corresponds to the highest *Il4* expression (Fig. 3e and Fig. S2c). Pseudotime analysis utilizing Monocle3 suggested the most mature cluster was cluster 6 which contained a substantial amount of Tfh-Ex cells (Fig. 3e and Fig. S2d). A small population of Tfh-Prog cells were also present in cluster 6, suggesting the possibility of a Tfh developmental pathway parallel to the IL-21 developmental pathways. Together these data suggest that both Tfh-Trans and Tfh-Ex cells have a phenotype existing between Tfh-Prog and Tfh-Full cells. To explore this aspect in more detail, we compiled a Tfh-Full versus Tfh-Prog gene module derived from differentially expressed genes in our bulk RNAseq dataset (from Fig. 1f) and used it to develop a feature score for each of the cells captured in the scRNAseq experiment. This feature score was highest in Tfh-Full cells, but Tfh-Trans and Tfh-Ex cells still had a higher score than Tfh-Prog cells which were similar to Tcon (Fig. 3f and Fig. S2b). These data suggest that Tfh-Ex cells largely maintain the Tfh-Full program even though they no longer express IL-21. Differential gene expression showed many genes differentially expressed between IL-21-produced subsets (Tfh-Trans, Tfh-Full, Tfh-Ex) and Tfh-Prog cells (Fig. 3g). 108 genes were similarly differentially expressed (DESeq2, $p < 0.05$) between all IL-21-experienced populations and Tfh-Prog cells including *Sh2d1a, Icos, Cd28,* and *Slamf6*. Average gene expression in Tcon and the four Tfh subsets showed that *Foxp1* was expressed in Tcon but peaked at the Tfh-Prog stage and was lowest in the Tfh-Full state (Fig. 3h). During the Tfh-Prog state *Batf* and *Icos* also begin to be expressed. However, other Tfh related genes such as *Bcl6* and *Cd40lg* do not begin to be expressed until the Tfh-Trans stage. All Tfh genes seem to peak during the Tfh-Full stage, and expression decreases during the transition to Tfh-Ex, although not back down to Tfh-Prog levels.

The expansion and distribution of TCR clones across Tfh subsets were also assessed at the single cell level by TCR immune profiling. Although many individual mice were combined in order to obtain enough cells for experiments, we nevertheless observed clonal expansion, particularly in clusters 2 and 6 (Fig. 3i and Fig S2e). Most clusters and cell states shared several overlapping clones, including Tfh-Ex (Fig. 3j). We also found robust shared clones between Tfh-Full cells and both Tfh-Trans and Tfh-Ex cells. When we compared the overlapping clones between Tfh-Trans or Tfh-Ex and Tfh-Full we found that overlapping clones were found for both substantially expanded and minimally expanded clones (Fig. 3k). Together, these data suggest four major developmental stages in Tfh cell differentiation which are transcriptionally distinct and once committed to an IL-21 producing stage, cells are unable to dedifferentiate to a Tfh-Prog state.

## Tfh developmental stage transitions are marked by epigenetic reorganization

To define the epigenetic relationships during the stepwise development of Tfh cells we compared the open chromatin landscape by ATAC-seq analyses. We sorted the four Tfh subsets (Tfh-prog, Tfh-Trans, Tfh-Full and Tfh-Ex) along with Tcon cells from dLNs of mice immunized with NP-OVA 9 days earlier (Fig. 4a). PCA of top variable peaks demonstrated clustering of IL-21-experienced (Tfh-Trans, Tfh-Full, Tfh-Ex) cells separately from Tfh-Prog and Tcon cells which clustered together (Fig. 4b). A similarity matrix demonstrated a higher correlation between IL-21-experienced Tfh subsets versus Tfh-Prog and Tcon cells (Fig. 4c). Moreover, the number of peaks opening or closing during the Tcon to Tfh-Prog transition were relatively few compared to the number of peaks opening/closing in the Tfh-Prog to Tfh-Trans or Tfh-Trans to Tfh-Full transitions (Fig. 4d, e). Interestingly, relatively fewer peaks were changed in the Tfh-Full to Tfh-Ex transition, consistent with a similar epigenetic landscape. These data suggest that most epigenetic remodeling associated with effector Tfh cells does not

occur until the Tfh-Trans stage, is strengthened during the Tfh-Full stage, and is largely maintained through the Tfh-Ex state. Although most epigenetic landscape changes occur in the Tfh-Trans stage, we could still detect some changes occurring in the Tfh-Prog stage. For instance, increased accessibility of the transcriptional start site (TSS) of *Cxcr5* occurs in Tfh-Prog cells and is maintained throughout all Tfh subset transitions, consistent with CXCR5 expression at the protein level (Fig. 4f). Similarly, two peaks in *Pdcd1* (PD-1) become accessible in Tfh-Prog cells which are maintained in all Tfh populations. Interestingly, one of these peaks which is ~23 kb upstream of the TSS has been implicated as being uniquely accessible in exhausted CD8+ T cells[30]. We next assessed the *Il21* gene locus to determine if the lack of IL-21 production in Tfh-Ex cells may be due to changes in chromatin accessibility. However, we did not find any peaks becoming inaccessible in Tfh-Ex cells. To the contrary, we found 6 distinct peaks (including one at the TSS) in or near the *Il21* locus that showed evidence of more accessibility starting in Tfh-Trans cells and were maintained even in Tfh-Ex cells. Therefore, diminished IL-21 production in Tfh-Ex cells was not due to altered chromatin accessibility of the *Il21* locus. In addition, other genes showed similar accessibility only starting in the Tfh-Trans stage. For instance, we found two peaks in the *Il4* locus which became accessible starting at the Tfh-Trans stage. We next determined the enrichment of transcription factor binding motifs in differentially accessible regions between subsets using Hypergeometric Optimization of Motif Enrichment (HOMER). We found that in Tfh-Prog versus Tcon cells there is increased accessibility ($p < 0.001$, fold change > 2) of DNA regions consistent with *Batf* and *Junb* binding sites (Fig. 4g). Additional enrichment for these binding motifs was found between Tfh-Full and Tfh-Prog cells, along with enrichment for *Tcf7, Bach2* and *Fos* binding sites. Likewise, we found significant enrichment of *Bach2* and *Junb* binding sites between Tfh-Full and Tfh-Trans and Tfh-Ex cells. Binding sites for *Lef1* were more accessible in Tfh-Full versus Tfh-Ex which was not found to be significant in other comparisons. Together, these data indicate the stepwise changes in chromatin accessibility that start to occur during the Tfh-Prog to Tfh-Trans stage transition, that reach their apex at the Tfh-Full stage, and are largely maintained in the Tfh-Ex stage.

## Tfh-Full and Ex stage cells promote GC stabilization and somatic hypermutation

Since Tfh cells might play distinct roles at the different stages of their differentiation, we sought to investigate the precise roles of later developmental stage Tfh cells in controlling germinal center responses. We developed a "Tfh-Full and Tfh-Ex DTR mouse" referred to as the F/Ex-DTR. The F/Ex-DTR combined the IL-21FM (*Il21*Cre*Rosa26*Lox-STOP-Lox-YFP) with a *Cxcr5*Lox-STOP-Lox-DTR allele. In the F/Ex-DTR mouse, Tfh-Full and Tfh-Ex cells can be potently and selectively deleted at any given time with administration of DT. We immunized F/Ex-DTR or control (*Il21*Cre*Rosa26*Lox-STOP-Lox-YFP*Cxcr5*wt) mice with an adjuvanted SARS-CoV-2 Spike protein vaccine, administered DT on days 2-11 to eliminate Tfh-Full and -Ex cells early during primary responses and harvested dLNs on day 14, which corresponds to the peak of the GC response (Fig. 5a). IL-21 experienced Tfh (Tfh-F/Ex) cells were potently deleted in F/Ex-DTR mice, as a frequency of total Tfh cells, as a frequency of CD4+ T cells and by microscopy (Fig. 5b, c and Fig. S3a). In contrast, CD4+CXCR5-YFP+ T cells did not decrease significantly. Deletion of Tfh-F/Ex cells resulted in a ~60% reduction in the frequency of GC B cells, suggesting that Tfh-F/Ex cells are essential for GC development (Fig. 5d). To understand the roles of Tfh-F/Ex cells in the GC response in more detail we used a recently developed single cell culture system to allow assessment of specificity on a clonal basis for GC B cells after SARS-CoV-2 vaccination (Fig. 5e)[5]. When we compared the GC B cell compartment in F/Ex-DTR and control mice using this system, we found deletion of Tfh-F/Ex cells did not change the relative frequency of

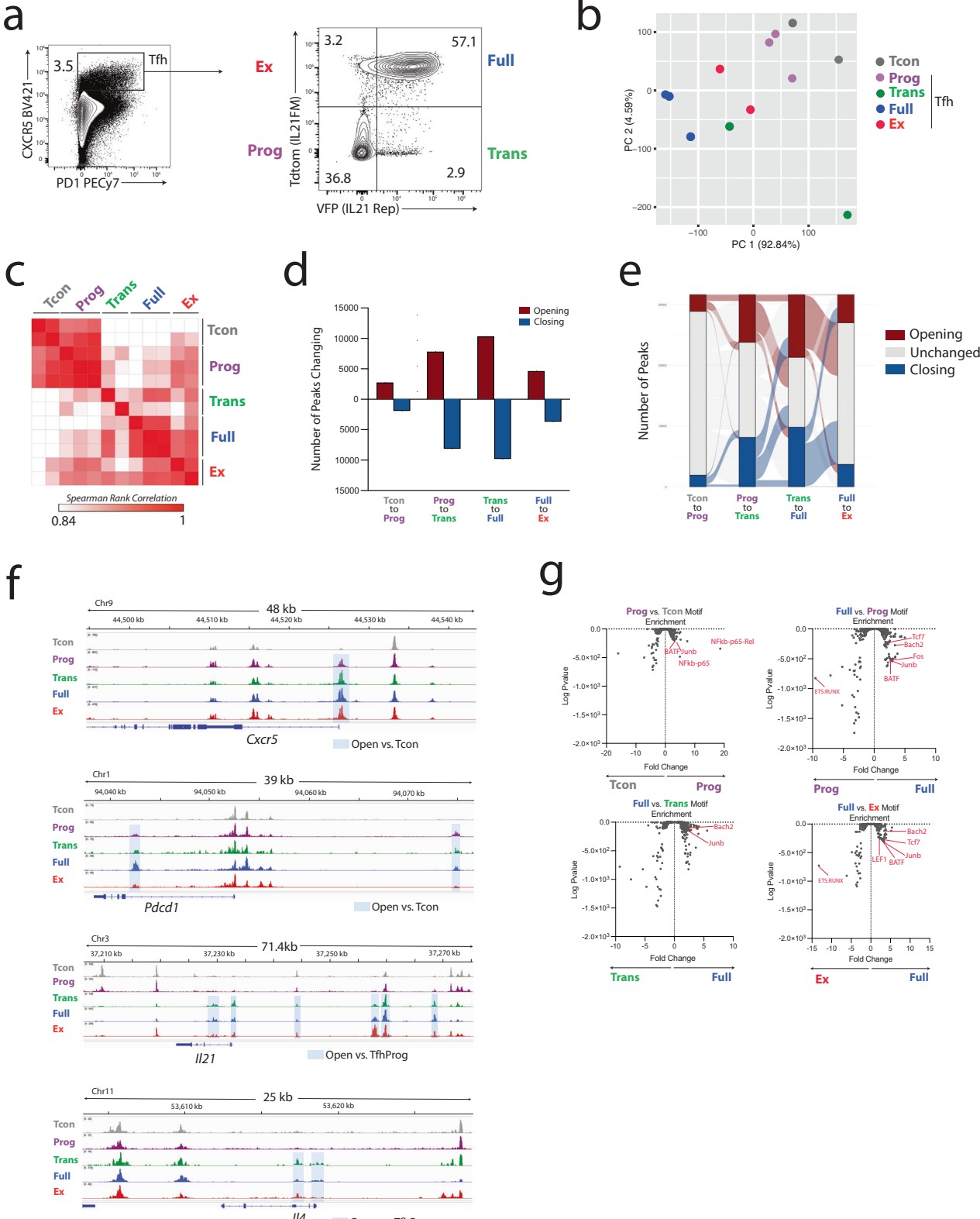

Spike-specific B cells in the IgG⁺ GC B cell pool, which was similar for both F/Ex-deleted (22.78%) and -sufficient mice (25.20%), even though the total frequency of GC B cells was much lower in DTR mice. However, when we mapped the specificity of GC B cells for individual domains in the Spike protein, we found that S1/RBD-specific clones dominated in control mice but were less than a third of the total in DTR mice. Notably, a consistent proportion of Spike clones from DTR mice could not be mapped to an individual domain, since they did not have reactivity to either S1 or S2 separately. This could be due to very low affinity BCRs or to the recognition of a conformational epitope by these clones. Together, these data suggest Tfh-F/Ex cells are essential for GC B cell development/expansion and may control vaccine-specific epitope dominance in the GC.

**Fig. 4 | Tfh developmental stage transitions undergo epigenetic remodeling.**
**a** Tg(*Il21*cre)*Rosa26*Lox-STOP-Lox-TdTomato*Il21*VFP mice were immunized with NP-OVA and
dLNs harvested on day 9. Tfh-Prog, Tfh-Trans, Tfh-Full, and Tfh-Ex (gated as shown)
as well as CD4+CXCR5–T conventional cells (Tcon) were sorted and processed for
Assay for Transposase-Accessible Chromatin with high-throughput sequencing
(ATAC-seq) analyses. Three replicates for the most abundant populations (Tfh-
Prog, Tfh-Full, and Tcon) and two for the least abundant (Tfh-Trans and Tfh-Ex)
were included in the analyses. DNA input was normalized during sequencing library
preparation. **b** Principal Component Analysis (PCA) of normalized ATACseq peak

counts. **c** Similarity matrix using Spearman Rank Correlation of ATACseq chromatin
accessibility data. **d** Number of peaks either opening or closing in indicated
developmental stage transitions. **e** Alluvial plot showing individual peaks opening,
closing or not changing throughout the Tfh developmental stage transitions. Only
peaks that change in at least one transition are shown. **f** Chromatin accessibility
tracks for Tfh developmental stage transitions at indicated loci of the genome. Blue
shaded regions are peaks differentially accessible compared to Tcon.
**g** Transcription factor motif enrichment analysis (HOMER) of peaks differentially
accessible in indicated Tfh developmental stages.

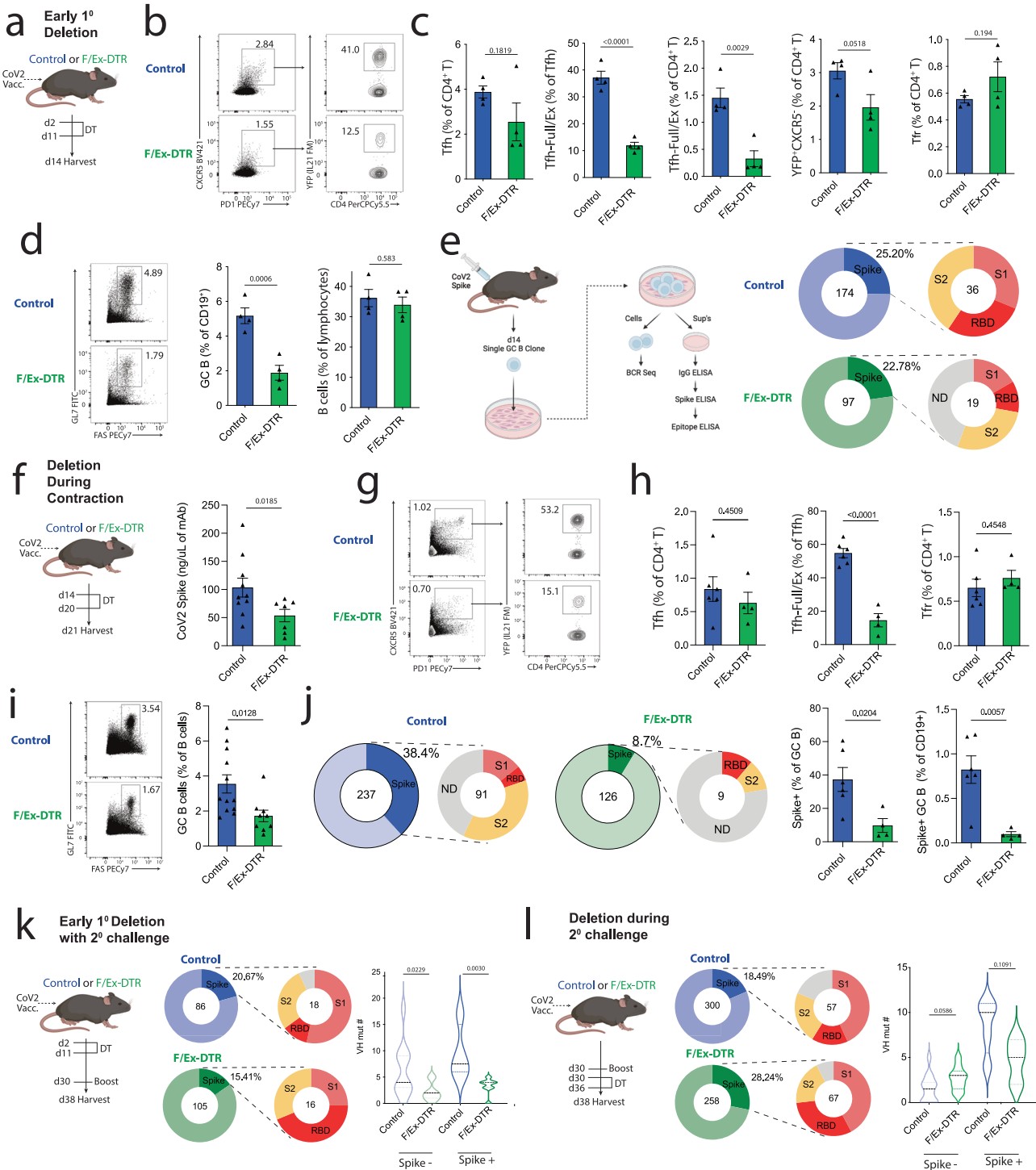

**Fig. 5 | Tfh-Full/Ex cells stabilize primary germinal centers to optimize somatic hypermutation after vaccine boosting. a** Control (*Il21*^Cre*Rosa26*^YFP*Cxcr5*^WT) or F/Ex-DTR (*Il21*^Cre*Rosa26*^YFP*Cxcr5*^LSL-DTR) mice were immunized with SARS-CoV-2 Spike and Tfh-Full/Ex deleted. **b** Gating of Tfh-Full and Tfh-Ex cells, pregated on CD4+CD19- cells. **c** Frequencies of Tfh-Full/Ex, Tcon21 (YFP+CXCR5-CD4+) and Tfr cells (*n* = 4). **d** Frequencies of GC or total B cells, pregated on CD19+ cells. (*n* = 4 Con, *n* = 5 DTR). **e** GC B single cell cultures (GC SCC) (left). Left pie charts indicate frequency of total Spike+ clones (of all IgG+). Numbers indicate total clones. Right pie charts indicate Spike domain specificity of Spike+ IgG+ clones. ND not detected. **f** Tfh-Full/Ex deletion during GC contraction. (right) Serology of SARS-CoV-2 Spike IgG (*n* = 10, Control; *n* = 17, DTR). Data are combined from 2 independent experiments, error bars indicate mean ± s.e.m.). **g** Gating of Tfh-Full/Ex cells, pregated on CD4+CD19-GITR-. **h** Frequencies of Tfh (CD4+CXCR5+PD1+GITR-), Tfh-Full/Ex (CD4+CXCR5+ PD1+GITR-YFP+), and Tfr (CD4+CXCR5+PD1+Foxp3+) cells (*n* = 6,

Control; *n* = 4, DTR). Error bars indicate mean ± s.e.m. **i** Frequency of GC B cells (*n* = 12, Control; *n* = 9, DTR), and is combined data from 3 independent experiments, error bars indicate mean ± s.e.m. **j** GC SCC assays. Data represents IgG+ clones in each group. Number indicates total clones (left). Bar graphs indicate the frequency of Spike+ GC B cells as a frequency of total GC B or total CD19+ cells per mouse. (*n* = 6, Control; *n* = 4, DTR), and is combined data from 3 independent experiments, error bars indicate mean ± s.e.m. **k** Deletion during primary GC followed by boosting and GC SCC. Clones are prescreened as IgG+. Number in circles represents the number of clones analyzed. Violin plots indicate distribution of VH mutations by *Igh* sequencing. (*n* = 11 Control (−); *n* = 4 Control (+); *n* = 9 DTR (−); *n* = 7, DTR (+)). **l** Schematic for deletion of Tfh-Full cells after boosting. Violin plots indicate distribution of VH mutations by *Igh* sequencing (*n* = 14 Control (−); *n* = 9 Control (+); *n* = 9 DTR (−); *n* = 3, DTR (+)). *P* values indicate unpaired two-tailed Student's *t* test (**c, d, h, i, j**) or Mann–Whitney test (**f, k, l**).

Since alterations in Tfh phenotype have been associated with GC resolution kinetics[8], we explored the role of Tfh-F/Ex cells during GC contraction. We vaccinated control or F/Ex-DTR mice with a SARS-CoV-2 Spike protein vaccine, administered DT starting at the peak of the GC response (on day 14) and harvested dLNs on day 21. At this time point, Spike-specific serum IgG levels were reduced to approximately half in the deleter mice (Fig. 5f). Despite effective deletion of Tfh-F/Ex cells in DTR compared to control mice, the frequency of total Tfh cells was not significantly different between the two groups (Fig. 5g, h). Consistent with the effect of Tfh-F/Ex observed at earlier time-points, GC B cells were significantly reduced in DTR compared to control mice (Fig. 5i). We performed single GC B cell culture assays and found that 38.40% of GC B cells were specific for SARS-CoV-2 Spike at day 21 in control mice (Fig. 5j). In contrast, only 8.70% of GC B cells were specific for SARS-CoV-2 Spike in the F/Ex-DTR mice at the same time point. The combined defects in GC B cells and decreases in Spike-specific cells in the GC compartment translated into substantially reduced frequencies of Spike-specific GC B cells in F/Ex-DTR mice. Taken together, these data indicate that Tfh-F/Ex cells mediate GC B cell differentiation and control epitope dominance during early GC responses and maintain vaccine-specific cells in the GC during GC contraction.

Next, we sought to determine the contribution of Tfh-F/Ex cells in primary GCs to recall responses after vaccine boosting. We vaccinated control or F/Ex-DTR mice with the adjuvanted SARS-CoV-2 Spike protein vaccine and deleted cells only from days 2 to 11 with DT. We boosted mice with another dose of the same SARS-CoV-2 Spike protein vaccine at day 30 and harvested dLNs 8 days after boosting (Fig. 5k). Discontinuing DT administration in this way allows loss of Tfh-F/Ex cells only during the primary GC response while leaving these cells unaffected during boosting. Consistently, deletion of Tfh-F/Ex during the primary GC response did not result in changes in total Tfh or IL-21 fate-mapped cells 8 days after vaccine boosting (Fig. S3b–c). Spike-specific IgG levels in the serum were marginally reduced in F/Ex-DTR compared to control mice but the difference was not statistically significant. Moreover, the frequency of GC B cells was similar between control and F/Ex-DTR mice after vaccine boosting (Fig. S3d–e). We next assessed the proportion of vaccine-specific cells within GCs after boosting and observed a slightly reduced (20.67% versus 15.41%) proportion of Spike-specific cells in GCs in DTR mice (Fig. 5k). However, RBD-specific clones were the most abundant Spike-specific clones in F/Ex-DTR mice and the least frequent Spike-specific clones in control mice. We sequenced the BCR heavy chain of both Spike-positive and -negative clones, and determined somatic hypermutation. We found reduced amounts of somatic hypermutation for both Spike-specific and non-specific clones in F/Ex-DTR compared to control mice (Fig. 5k). These data suggest that Tfh-F/Ex roles during the primary GC reaction contribute to optimal somatic hypermutation after vaccine boosting.

To determine the role of Tfh-F/Ex cells specifically during vaccine boosting, we vaccinated control or F/Ex-DTR mice with the SARS-CoV-

2 Spike protein vaccine, boosted mice on day 30, administered DT to delete Tfh-F/Ex cells, and harvested dLNs 8 days after boosting (Fig. 5l and Fig. S3f–g). The frequency of GC B cells was slightly attenuated in DTR compared to control mice (Fig. S3h). We also performed single GC B cell cultures but surprisingly found an increased percentage of Spike-specific clones in the GC pool of DTR mice with roughly the same distribution of epitope specificity (Fig. 5l). We sequenced the BCR heavy chain from Spike-specific and non-specific B cell clones from these assays and found the amount of somatic hypermutation was slightly reduced in Spike-specific GC B cell clones, although this did not reach statistical significance. Together, these data indicate that IL-21 fate-mapped Tfh cells have potent early roles in GCs to promote somatic hypermutation after vaccine boosting, whereas during secondary GCs Tfh-F/Ex have only minor roles and may even disproportionately promote non-antigen-specific clones in GCs.

### In vivo progenitor capabilities of Tfh developmental stages

Our finding that IL-21 fate-mapped Tfh subsets (Full, Ex) have essential roles in modulating GC responses inspired us to assess possible functions of the Tfh-Prog population. Studies in chronic LCMV infection as well as in Th17 cells suggest some T cell progenitor populations can function to provide uncommitted cells that rapidly develop into effector cells[31,32]. To determine if Tfh-Prog can perform similar functions we performed adoptive transfer assays. In these assays we transferred CD4+ T cells from OT-II+ IL-21^FM/Rep mice to WT recipients, which were immunized with NP-OVA. After 9 days, Tfh subsets were sorted and transferred to immunized *Tcra*^−/− recipients which were then harvested 12 days later to assess expansion and further differentiation of Tfh subsets (Fig. 6a). Tfh-Prog transferred cells differentiated into multiple populations including Tfh-Full (Fig. 6b). Importantly, Tfh-Prog cells gave rise to a larger number of total Tfh cells compared to other Tfh-subsets suggesting progenitor potential, although data were quite variable likely due to differences in prior expansion of the Tfh-Prog before transfer (Fig. 6c). Importantly, the expansion of Tfh-Prog also corresponded to development into Tfh-Full cells as determined by increased VFP expression. Interestingly, a population of Tfh-Full lost IL-21 expression to become Tfh-Ex, and sizable population of Tfh-Ex regained IL-21 expression in vivo, confirming results observed in vitro (Fig. S4a–d). To understand if the increased expansion of Tfh-Prog into effector Tfh-Full cells resulted in increased B cell responses we measured GC B cell differentiation. Recipients of Tfh-Prog transferred cells had the highest frequencies of both GC B cells as well as CD138+ plasma cells (Fig. 6d, e). Together, these data suggest that Tfh-Prog can function as a progenitor population that can give rise to a robust population of Tfh-Full cells, which promote B cell responses.

### Foxp1 regulates developmental transitions in Tfh cells

In order to evaluate possible intrinsic factors controlling the differentiation of Tfh-Prog to Tfh-Full cells, we focused on transcription

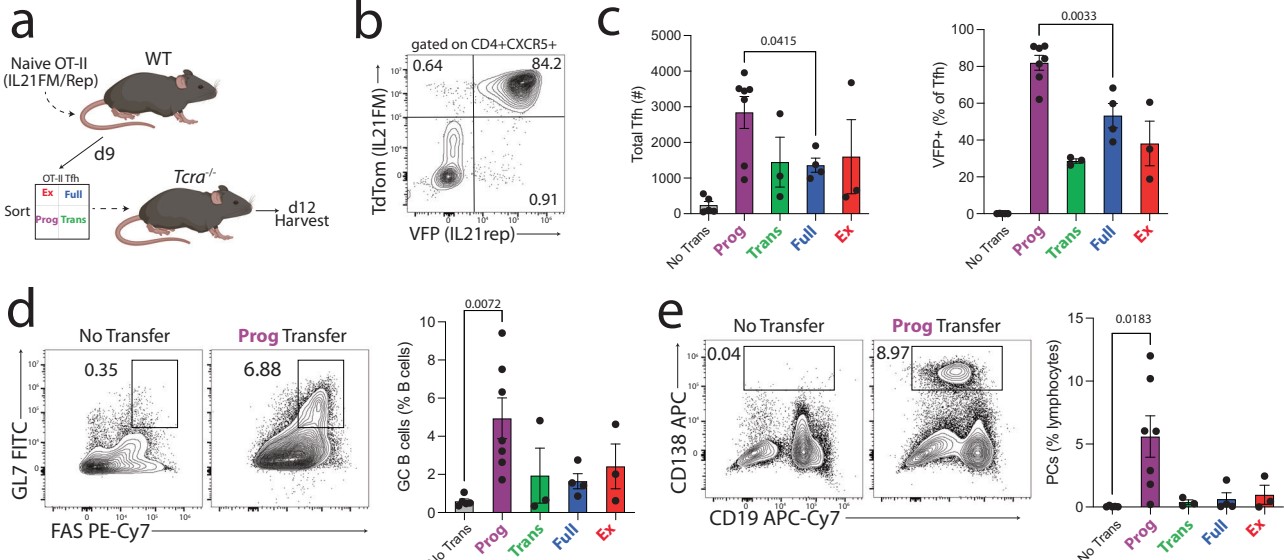

**Fig. 6 | In vivo progenitor capabilities of Tfh developmental stages. a** Diagram of experiment. CD4 T cells from naive OT-II+Tg (*Il21*cre)*Rosa26*Lox-STOP-Lox-TdTomato*Il21*VFP were transferred to WT mice which were immunized with NP-OVA. On day 9 Tfh subsets were sorted and individually transferred to immunized *Tcra*-/- recipients. dLNs were harvested 12 days later. **b** Representative gating identifying Tfh developmental stages in transferred cells. Representative plots derive from concatenated replicates and are pregated on CD4+CXCR5+ cells. **c** Total numbers of Tfh cells (left) and frequency of VFP+ cells within Tfh cells (right). *X*-axis indicates the individual population that was transferred to each recipient mouse (*n* = 4 No Transfer, *n* = 7 Prog, *n* = 3 Trans, *n* = 4 Full, *n* = 3 Ex). Error bars indicate mean ±

s.e.m. *P* value was calculated using unpaired two-tailed Student's *t* test. **d** Gating strategy (left) and frequency (right) of GC B cells from transfer experiments as in (**a**). (*n* = 5 No Transfer, *n* = 7 Prog, *n* = 3 Trans, *n* = 4 Full, *n* = 3 Ex). Error bars indicate mean ± s.e.m. *P* value was calculated using unpaired two-tailed Student's *t* test. **e** Gating strategy (left) and frequency (right) of plasmablast/plasma cells (PCs) from transfer experiments as in (**a**). (*n* = 5 No Trans, *n* = 7 Prog, *n* = 3 Trans, *n* = 4 Full, *n* = 3 Ex). Error bars indicate mean ± s.e.m. *P* value was calculated using unpaired two-tailed Student's *t* test. Data are combined data from two separate experiments.

---

factors expressed at the Tfh-Prog stage that were attenuated in the Tfh-Trans through Tfh-Ex stages. Among candidate genes fitting these criteria, we focused on the transcription factor FoxP1. FoxP1 has previously been implicated as a negative regulator in initial Tfh differentiation from naïve T cells, as well as in chromatin remodeling in Treg cells[26,33]. To assess the possible role of FoxP1 in controlling the Tfh-Prog to Tfh-Full transition, we crossed IL-21 reporter (*Il21*VFP) and tamoxifen-inducible FoxP1 knockout (UBCCreERT2*Foxp1*fl/fl) mice, so that we could sort Tfh-Prog cells, delete FoxP1 inducibly, and assess synchronized de novo Tfh-Full differentiation. We immunized control (*Il21*VFP*Foxp1*wt) or iFoxp1Δ (*Il21*VFPUBCCreERT2*Foxp1*fl/fl) mice with NP-OVA and administered Tamoxifen to start the *Foxp1* allele excision process. After 7 days, we sorted Tfh-Prog cells (CD4+CXCR5+VFP−) from both groups of mice and cultured them with CD45.1+ B cells and CD45.1+ bystander Tfh cells for 4 days in the presence of anti-IgM and anti-CD3/CD28 beads (Fig. 7a). Cultures were then harvested and the frequency of Tfh-Full de novo differentiation was assessed. We found only a small fraction of Tfh-Full de novo differentiation from control Tfh-Prog cells, consistent with experiments in Fig. 2g (Fig. 7b). Tfh-Full de novo differentiation was substantially increased in FoxP1-deleted Tfh-Prog cells suggesting FoxP1 is a potent positive regulator of the Tfh-Prog stage and limits further development to the Tfh-Full stage. Interestingly, we also found that de novo differentiated Tfh-Full cells displayed higher IL-21 expression on a per cell basis in FoxP1-deleted cells (Fig. 7c). ICOS and PD-1 expression was also higher in Tfh-Full cells from FoxP1-deleted conditions suggesting a Tfh-Full program beyond IL-21 expression (Fig. 7c and Fig. S5a–f). Bulk RNAseq transcriptional analysis on Tfh-Full from control or Foxp1-deleted conditions in vitro revealed no meaningful differences in Tfh-Full genes yet enrichment for genes involved in cellular processes and E2F, ZF5, and FoxN4 transcription factor motifs, despite high variability due to a lower number of replicates (Fig. S5g–h).

To further understand how FoxP1 controls the Prog to Full transition, we administered Tamoxifen to immunized control or iFoxp1Δ mice, sorted Tfh-Prog or Tfh-Full cells and performed direct ex vivo bulk RNAseq transcriptional analysis. The number of differentially expressed genes with FoxP1 deletion was higher in Tfh-Prog (505 genes, adjusted *p* < 0.05) compared to Tfh-Full cells (37 genes, adjusted *p* < 0.05), consistent with FoxP1 being more highly expressed during the Tfh-Prog stage and lowest during the Tfh-Full stage (Fig. 7d). Of the 505 DEGs in control versus FoxP1-deleted Tfh-Prog cells, 97 have been shown to be direct targets of FoxP1 utilizing available ChIP-Seq datasets[33]. Pathway analysis utilizing gProfiler indicated enrichment for transcription factor binding motifs including ZF5, E2F family members, FoxN4 and Bcl6b in DEGs between control and FoxP1-deleted TfhProg cells (Fig. S6a). To confirm that FoxP1-deleted Tfh-Full cells were bona fide Tfh-Full cells, we assessed the expression of the Tfh-Full gene module by gene set enrichment analysis (Fig. S6b). We found minimal differences between control and FoxP1-deleted Tfh-Full, suggesting that loss of FoxP1 in Tfh-Prog cells resulted in further development into Tfh-Full cells that were transcriptionally similar to control Tfh-Full cells. We assessed the overlap of DEGs from FoxP1-deleted Tfh-Prog and Tfh-Full, and further compared them with FoxP1 target genes (Fig. S6c, d). Only two FoxP1 target genes, *Cep70* and *Fhit* were DEGs in both Tfh-Prog and Tfh-Full suggesting that FoxP1 likely regulates transcriptional programming differently in Tfh-Prog versus Tfh-Full. At the protein level IL4R, but not Ly6a, was more highly expressed in FoxP1-deleted TfhProg and Tfh-Full compared to control cells (Fig. 7e).

Since we found that FoxP1 is upregulated during the Tfh-Ex stage, we also determined whether FoxP1 could regulate the Tfh-Full to Tfh-Ex transition. We performed in vitro experiments using Tfh-Full cells as an input and assessed the frequency of conversion to Tfh-Ex cells via downregulation of VFP expression. We found that a proportion of WT

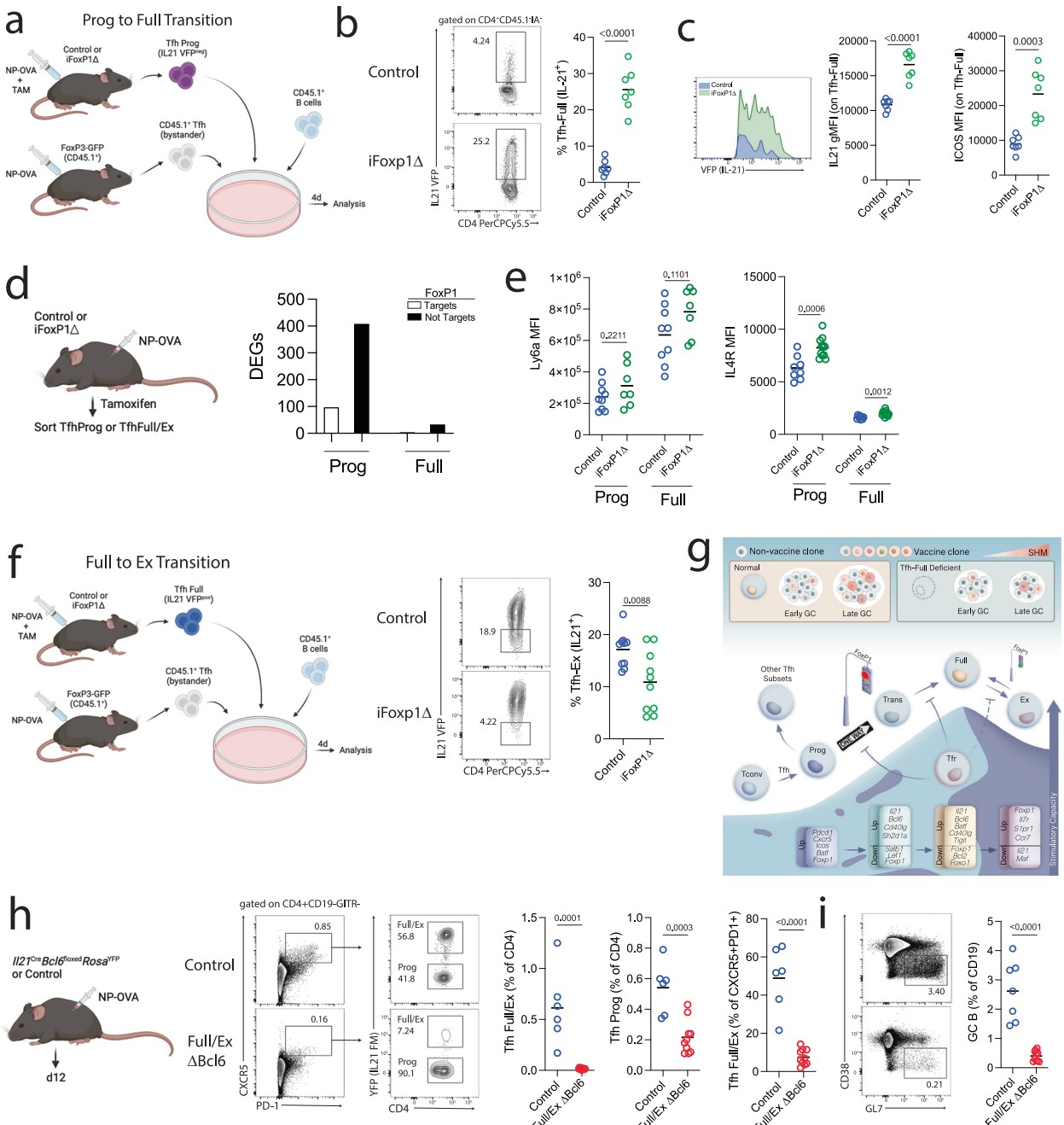

**Fig. 7 | Foxp1 regulates Tfh developmental stage transitions. a** Schematic of Tfh-Prog to Tfh-Full transition assay. Control (*Il21*^VFP*Foxp1*^wt) or iFoxp1Δ (*Il21*^VFP*Foxp1*^fl/fl*UBC*^CreERT2) mice were immunized and tamoxifen given to start Foxp1 deletion. On day 7 Tfh-Prog (CD4⁺CXCR5⁺GITR^negVFP^neg) cells were cultured with CD45.1⁺ bystander Tfh (CD4⁺CXCR5⁺GITR^neg) and B (CD19⁺) cells for 4 days with anti-IgM and anti-CD3/CD28 beads. **b** Identification of de novo Tfh-Full cells by identifying CD4⁺VFP⁺ cells (left) and frequency of Tfh-full cells (right) *n* = 7 replicate cultures per group. Line indicates mean. Data are from one experiment and is representative of two independent experiments. *P* value was calculated using unpaired two-tailed Student's *t* test. **c** IL-21 expression (left) and ICOS (right) levels on Tfh-full cells. Data are from one experiment and is representative of two independent experiments. *n* = 7 replicate cultures per group. Line indicates mean. *P* value was calculated using unpaired two-tailed Student's *t* test. **d** Ex vivo transcriptional analysis of Tfh-Prog and Tfh-Full by bulk RNASeq. *n* = 4 replicates per group. DEGs = *P* < 0.05 by EdgeR. **e** Surface expression of Ly6a or IL4R on ex vivo TfhProg or TfhFull. *n* = 9 (Control) or *n* = 7 (DTR). *P* value was calculated

using unpaired two-tailed Student's *t* test. **f** Tfh-Full to Tfh-Ex transition assay. Control (*Il21*^VFP*Foxp1*^wt) or iFoxp1Δ (*Il21*^VFP*Foxp1*^fl/fl*UBC*^CreERT2) mice were immunized and tamoxifen given to delete FoxP1. On day 7 Tfh-Full (CD4⁺CXCR5⁺GITR^negVFP⁺) were cultured with CD45.1⁺ bystander Tfh (CD4⁺CXCR5⁺GITR^neg) and B (CD19⁺) cells for 4 days. De novo Tfh-Ex cells were identified as CD4⁺VFP⁻ cells. *n* = 10 replicate cultures per group. Data are concatenated data from two independent experiments. *P* value was calculated using unpaired two-tailed Student's *t* test. **g** Schematic of Tfh developmental trajectories and individual functions of stages. **h** Deletion of Bcl6 in the Tfh-Full stage. F/ExΔBcl6 (*Il21*^Cre*Bcl6*^floxed*Rosa*^YFP) or control (*Il21*^Cre*Bcl6*^wt*Rosa*^YFP) mice were immunized and harvested on day 12. Tfh-Full/Ex (CD4⁺CXCR5⁺PD1⁺GITR^negYFP⁺) or Tfh-Prog (CD4⁺CXCR5⁺ PD1⁺GITR^negYFP⁻) were assessed. *n* = 6 (Control) or *n* = 10 (ΔBcl6). Data are concatenated from two independent experiments. *P* value was calculated using unpaired two-tailed Student's *t* test. **i** Frequency of GC B cells from mice as in (**h**). *P* value was calculated using unpaired two-tailed Student's *t* test.

Tfh-Full cells downregulated VFP to become Tfh-Ex cells, and this proportion was attenuated in FoxP1-deleted conditions (Fig. 7f). This suggests that FoxP1 is also a positive regulator of the Tfh-Ex stage (Fig. 7g).

Our findings indicate that FoxP1 has a more substantial role in Tfh-Prog versus Tfh-Full cells, that it has a reciprocal expression pattern compared to *Bcl6*, that deletion of FoxP1 in Tfh-Prog results in alteration in genes with Bcl6b binding motifs, and Bcl6 binding sites have overlap with E2F[34]. We therefore hypothesized that FoxP1 may partially function by altering downstream Bcl6-regulated pathways, in a manner similar to Blimp1. In support of this, we found changes in seven Bcl6 target genes in FoxP1-deleted Tfh-Prog, utilizing a previously published Bcl6 ChipSeq dataset (Fig. S6d)[35]. These data suggest that Bcl6 signaling may be essential to form and stabilize the Tfh-Full stage through reciprocal modulation with FoxP1.

Bcl6, the master transcription factor for Tfh cells, has previously been shown to be important in the maintenance of Tfh cells over time[36–38]. However, the continued role of Bcl6 in specific Tfh subsets, such as Tfh-Full cells, has not been fully studied. To test the roles of Bcl6 specifically during the Tfh-Full stage, we immunized Full/ExΔBcl6 (*Il21*[Cre]*Bcl6*[floxed]*Rosa*[YFP]) or control (*Il21*[Cre]*Bcl6*[wt]*Rosa*[YFP]) mice and assessed the frequency of TfhFull/Ex cells. We found profound decreases in Tfh-Full/Ex cells, along with less substantial decreases in Tfh-Prog, translating into Tfh-Full/Ex being almost absent from the Tfh compartment (Fig. 7h). The frequency of GC B cells was also substantially reduced in Full/ExΔBcl6 compared to control mice (Fig. 7i). Together these data demonstrate the essential role of Bcl6 in maintaining the Tfh-Full stage, the reciprocal relationship between FoxP1 and Bcl6 in controlling Tfh fate decisions and how altering FoxP1 or Bcl6 signals can regulate Tfh stage transitions and stability.

## Discussion

Tfh cell states have been classically divided into pre-Tfh and GC-Tfh cells. However, this nomenclature has been recently challenged because GC-Tfh cells are too numerous to be localized only in GCs, and newer studies suggest localization outside of these areas[7]. Moreover, although the prototypical Tfh cytokine IL-21 has been thought to be produced mainly in GCs, it can be produced in other anatomical locations, e.g. by circulating Tfh cells. Here we developed an anatomical location agnostic strategy to uncover the developmental relationships of Tfh cells and found four sequential stages (Tfh-Prog, Tfh-Trans, Tfh-Full, and Tfh-Ex) which are transcriptionally and epigenetically distinct. The Tfh program is largely induced starting at the Tfh-Trans stage and is strengthened in the fully developed Tfh-Full stage. Tfh-Ex cells maintain key transcriptional and epigenetic programming but lose the ability to effectively produce IL-21. Extrinsically, Tfr cells regulate progression through these developmental stages to control humoral immunity. Intrinsically, Foxp1 acts as a checkpoint to both insulate the Tfh-Prog state as well as accelerate conversion of Tfh-Full to Tfh-Ex cells. Therefore, both intrinsic and extrinsic factors balance the progenitor-to-effector ratio to optimize immunity. We also show that Tfh-Full/Ex cells have multifaceted functional roles in controlling humoral immunity, including promoting initial GC formation and vaccine epitope dominance, preventing GC contraction, and ensuring maximal somatic hypermutation after vaccine boosting. Importantly, although Tfh-Full are enriched in GCs, all four developmental stages can be found in GCs and in B cell follicles. Together with chromatin accessibility changes, these data point to developmental stage progression occurring largely independently of anatomical location.

The Tfh-Prog state is marked by increases in CXCR5, PD1, BATF and ICOS, but is transcriptionally and epigenetically closer to conventional T cells than other Tfh subsets. Therefore, the Tfh-Prog state is similar in principle to "pre-Tfh" cells. The Tfh-Trans stage, which is relatively short-lived, is marked by substantial transcriptional and epigenetic reprogramming that further occurs during the transition to the Tfh-Full stage. The Tfh-Full stage is marked by the highest expression of Tfh-related genes, many of which overlap with genes predicted to be highly expressed on the most mature Tfh cells in human lymph nodes[39]. Although the Tfh-Full cells we identify are the most differentiated stage, it is possible that some heterogeneity may exist in this population and that a subset may be more differentiated (e.g. PD-1 or Bcl6[hi]) than others within this population. However, fate mapping these populations is not possible with current tools. Nevertheless, our data indicate that Bcl6 is required during the Tfh-Full stage to maintain these cells. This is consistent with previous data showing that Bcl6 was required for the maintenance of total Tfh effector and/or memory cells[36–38]. Some Tfh-Full extinguish IL-21 production to become Tfh-Ex cells, a stage that largely maintains epigenetic remodeling of Tfh-Full cells but is transcriptionally similar to Tfh-Trans cells. Although we identify FoxP1 as a factor that controls Tfh-Ex development, other signals such as loss of antigenic signals and/or costimulation may also be involved and are not mutually exclusive. The maintenance of a similar transcriptional/epigenetic reprogramming by all IL-21 fate-mapped Tfh subsets suggests that differentiation from Tfh-Prog cells is a one-way transition, and that Tfh-Full cells cannot dedifferentiate back to the Tfh-Prog state. This makes insulation of the progenitor state essential. We show that Tfh-Prog cells exist in a poised, but not committed progenitor-like state that can give rise to substantial amounts of Tfh-Full cells in vivo. These data suggest Tfh-Prog have the most stem-like capabilities of all the Tfh developmental stages tested. However, it is also possible that Tfh-Prog may differentiate into other types of effector Tfh cells in a parallel developmental pathway to the IL-21 dependent pathway. Alternatively, these cells may have unique functions in humoral immunity such as formation of memory cells.

To assess the function of IL-21 experienced Tfh subsets, we generated a F/Ex-DTR mouse which deletes Tfh-Full and Tfh-Ex cells in an inducible manner. We found multifaceted roles for these cells in the context of SARS-CoV-2 vaccination including GC formation, prevention of GC contraction, as well as control of vaccine epitope dominance and somatic hypermutation after vaccine boosting. These data suggest that IL-21 fate-mapped Tfh subsets are the most functional to support humoral immunity, at least during primary GC responses. The loss of IL-21 expression but maintenance of the larger Tfh-Full program in the Tfh-Ex stage suggests that these cells may support the GC response in IL-21 independent ways or, alternatively, may promote GC contraction. Recently, FoxP3 expression has been shown to occur in a small subset of Tfh cells during late GC responses and these cells have been predicted to promote GC contraction[8]. Unlike these FoxP3[+] Tfh cells, Tfh-Ex (of which only ~3% express FoxP3) cells arise much earlier during the GC reaction and maintain expression of genes important for B cell help, such as *Cd40lg*. Nevertheless, both populations express little IL-21 and may mediate contraction by diverting resources away from Tfh-Full cells. Our finding that Tfh-Ex cells can in some settings regain IL-21 expression to become Tfh-Full cells suggests that Tfh-Ex may also exist as a reservoir of cells without full progenitor potential that can be quickly converted to Tfh-Full cells to maintain the needs of the GC reaction. In contrast, Tfh-Prog can differentiate into Tfh-Full cells during ongoing GC responses, but this process may take much longer due to the sequential epigenetic remodeling and transcriptional programming that is necessary. Since our F/Ex-DTR mouse does not delete Tfh-Prog or Tfh-Trans cells which can also be present in GCs, the strong phenotype in GC reduction in F/Ex-DTR suggest Tfh-Prog/Trans have more limited direct stimulatory capacity but may have other roles yet to be uncovered.

Since Tfh-Full cells have potent roles in vaccine-specific antibody responses, a strategy to enhance vaccine efficacy may be to promote the transition from Tfh-Prog cells to more functional Tfh-Full cells. Alternatively, it may be possible to insulate Tfh-Full cells from further differentiation to Tfh-Ex cells. We found that Foxp1 is a regulator of Tfh

developmental transitions, particularly in the Tfh-Prog state. Deletion of Foxp1 specifically in Tfh-Prog cells leads to enhanced differentiation to Tfh-Full cells. Therefore, Foxp1 may help to maintain an uncommitted progenitor population of Tfh-Prog cells. Interestingly, although Foxp1 transcripts are reduced during the Tfh-Full stage, they are not as attenuated in the Tfh-Ex stage. Our finding that Foxp1 accelerates the conversion of Tfh-Full to Tfh-Ex cells suggests that Foxp1 may act as a rheostat and control the precise composition of progenitor, fully differentiated, and Ex cells to optimize immunity. The factors that control Foxp1 expression during the distinct Tfh developmental stages are still unclear. However, the local availability of antigen to drive TCR signals as well as costimulation may be key mediators. In addition, how Foxp1 controls Tfh stage progression is still unclear. The lack of substantial transcriptional differences in FoxP1-deleted Tfh-Full cells suggest that FoxP1 likely insulates the Tfh-Prog state to prevent further development. Although our data suggest that FoxP1 deletion results in alterations in some downstream Bcl6 pathways, it is unclear if FoxP1 regulates stages only by controlling Bcl6-regulated genes. Studying this in more depth is difficult due to the inability to perturb Foxp1 without causing Tfh-Full development. Nevertheless, the profound loss of Tfh-Full with Bcl6 deletion suggest that Bcl6 is essential to maintain Tfh-Full cells. Therefore, dampening of Bcl6-mediated pathways by FoxP1 would be predicted to maintain Tfh-Prog cells, similar to how Blimp1 reciprocally modulates Bcl6 during initial Tfh development. Together, our data demonstrate previously unappreciated developmental stages in Tfh cells. By understanding the functions and regulation of these developmental stages, new strategies can be developed to enhance vaccine and anti-viral immunity or to limit autoimmunity by fine tuning the developmental stages of Tfh cells.

## Methods

### Mice
All animals were used according to Brigham and Women's Hospital Institutional Animal Care and Use Committee policies, as well as the National Institute of Health guidelines. $Rosa26^{Lox-STOP-Lox-YFP}$, $Rosa26^{Lox-STOP-Lox-TdTomato}$, $Il21^{VFP}$, $Foxp3^{IRES-Cre-YFP}$, $Ptprc^a$, $UBC^{Cre-ERT2}$, OT-II, $Bcl6^{fl/fl}$ and $Foxp1^{fl/fl}$ mice, all on the C57BL/6 background, were purchased from Jackson Laboratories. $Tg(Il21^{Cre})$ were a kind gift from Uta Hoepken and has been published previously[19]. $Foxp3^{IRES-GFP}$ and $Cxcr5^{IRES-LoxP-STOP-LoxP-DTR}$ mice have been published previously[6,40]. Males and females were included in the study. All mice were 6-10 weeks old. Mice were kept at a 12/12 h dark/light cycle, at 22 °C and 42% humidity Mice were fed 5053 PicoLab rodent diet 20 (LabDiet).

### Immunizations and treatments
Mice were immunized with 100 µg NP-OVA (Biosearch Technologies) emulsified (1:1) in complete Freund adjuvant (CFA, Sigma-Aldrich) or mixed in Addavax adjuvant (Invivogen) for subcutaneous immunization on both flanks or emulsified in incomplete Freund adjuvant (IFA, Sigma-Aldrich) for intraperitoneal immunization. For SARS-CoV-2 studies, mice were immunized subcutaneously on the flank with 5 µg of 2019-nCoV Spike Protein (Sino Biological) mixed (1:1) with Addavax adjuvant. For prime-boost experiments, mice received a boost dose with the same composition 30 days after the primary immunization. For deletion experiments, mice were intraperitoneally administered with 0.5 µg of diphtheria toxin in PBS at indicated timepoints. In experiments with mice bearing the $UBC^{Cre-ERT2}$ allele, Tamoxifen (1.5 mg/dose) was injected intraperitoneally for five consecutive days from the day of immunization to induce the activity of Cre recombinase, and mice were euthanized after seven days from the first Tamoxifen dose.

### Flow cytometry
Single cell suspensions were incubated with the following antibodies directed against surface markers for 30 min at 4 °C (1:200 dilution

unless otherwise specified): anti-CD4 (Biolegend, RM4-5 or BD, GK1.5), anti-CD19 (Biolegend, 6D5 or BD, 1D3), anti-ICOS (Biolegend, 15F9), anti-PD-1 (Biolegend, RMP1-30), anti-CXCR5 biotin (Biolegend, L138D7), anti-GITR (Biolegend, DTA-1), anti-CD45.1 (Biolegend, A20), anti-CD124 (BD Pharmingen, mIL4R-M1), anti-Ly6a (Biolegend, clone D7), anti- T- and B-cell activation antigen (BD Biosciences, GL-7), anti-FAS (BD, JO2), anti-CD38 (Biolegend, 90), anti-IA/IE (BD, 1:1000, 2G9). Cells were subsequently stained with Streptavidin-BV421 (Biolegend, 1:200, #405225) for 20 min at 4 °C for CXCR5 detection. For intracellular staining, samples were fixed with the Foxp3 Fix/Perm buffer set according to the manufacturer's instructions (eBioscience) and stained with anti-FoxP3 (eBiosciences, FJK-16s). Stained single cell suspension was acquired on a Cytek AURORA (5-laser configuration), and data were further analyzed with FlowJo version 10.

### Sorting
Draining lymph nodes or spleens were mashed through 70-µm filters and resuspended in PBS supplemented with 1% FBS and 1 mM EDTA. CD4$^+$ T cells were enriched by magnetic positive selection according to the manufacturer protocol (Miltenyi Biotec). CD4-enriched T cells and CD4-depleted cell suspension were then stained and sorted on a BD FACS Aria II cell sorter (85 µm nozzle) or on a CytoFLEX SRT sorter using optimal purity settings.

### In vitro Tfh differentiation assays
Seven days post-immunization, mice were sacrificed and either draining lymph nodes or spleen were harvested. Single cell suspensions were stained with surface antibodies and sorted according to the experimental setup. Sorted populations comprised Tfh-Prog (CD45.1$^-$CD4$^+$CD19$^-$CXCR5$^+$GITR$^-$IL21$^-$), Tfh-Full (CD45.1$^-$CD4$^+$CD19$^-$CXCR5$^+$GITR$^-$IL21$^+$), Tfr (CD45.1$^+$CD4$^+$CD19$^-$CXCR5$^+$FoxP3$^+$), bystander Tfh (CD45.1$^+$CD4$^+$CD19$^-$CXCR5$^+$FoxP3$^-$) and B cells (CD45.1$^+$CD19$^+$CD4$^-$). Cells were resuspended in R10 medium (RPMI 1640, 10% FBS, 10 mM HEPES, 100 U/mL Penicillin, 100 µg/mL Streptomycin, 55 µM β2-ME) and seeded in a total volume of 50 µL in sterile 384-well round bottom plates (Thermo Fisher Scientific), at a concentration of 250 and 150 cells/µL for B cells and Tfh/Tfr cells, respectively. Each well was supplemented with soluble anti-IgM (Jackson ImmunoResearch, 5 µg/mL) and either soluble anti-CD3 (Bioxcell, 2C11, 2 µg/mL) or anti-CD3/anti-CD28 DynaBeads (Gibco) according to manufacturer's instructions. After 4 days of culture at 37 °C with 5% CO$_2$, cells were harvested, stained with fluorochrome-labeled antibodies, and analyzed.

### Single-cell GC B cell cultures
Lymph node single-cell suspensions were prepared as described above and incubated with the following antibodies for 30 min at 4 °C (1:200 dilution): anti-B220 (Biolegend, RA3-6B2), anti-CD38 (Biolegend, Clone 90), T- and B-cell activation antigen (BD Biosciences, GL-7), anti-CD138 (Biolegend, 281-2) and anti-CD4 (Biolegend, RM4-5). Single GC B cell cultures were performed as described previously[5]. Briefly, single cells were sorted into 96-well round-bottom plates (Corning) containing $1 \times 10^3$ NB21.2D9 cells, a kind gift from Dr. Garnett Kelsoe. Cells were cultured in OptiMEM (GIBCO) supplemented with 10% heat-inactivated fetal bovine serum (GIBCO, 16140071), 2 mM L-glutamine, 1 mM sodium pyruvate, 50 µM 2-mercaptoethanol, 100 U penicillin, and 100 µg/ml streptomycin. IgG secretion was assessed by ELISA after 6 days of culture and IgG+ supernatants were collected after 9 days of culture. Cells were frozen in TCL lysis buffer supplemented with 1% 2-mercaptoethanol for $Igh$ sequencing.

### ELISA
To determine serum or single GC B cell culture supernatant levels of IgG antibodies against the SARS-CoV-2 spike protein and its subunits, Maxisorp (Nunc) plates were coated overnight with 1 µg/mL 2019-nCoV Spike Protein (Sino Biological), 2 µg/mL 2019-nCoV Spike S1-His (Sino

Biological), 2 μg/mL 2019-nCoV Spike S2-His (Sino Biological), or 2 μg/mL 2019-nCoV Spike RBD-His Protein (ABclonal). 96-well half area plates (Microlon, Greiner) were used for supernatants with working volumes reduced by half. Plates were blocked for one hour with 1% BSA at 37 °C, and serum sample dilutions or undiluted cell culture supernatant were incubated for one hour at room temperature. After washing, an alkaline phosphatase-conjugated anti-IgG secondary antibody (Southern Biotech, 1:1000 dilution) was incubated for one hour, and plates were developed with phosphatase substrate (Sigma) dissolved in Pierce substrate buffer (Thermo Fisher Scientific). Optical density was measured with a plate reader (Spectramax).

## Bulk RNA sequencing

Bulk RNA-seq was performed as described previously[6,18]. Briefly, RNA was isolated using MyOne Silane Dynabeads (Thermo Fisher Scientific). RNA was fragmented and barcoded using 8 bp barcodes in conjunction with standard Illumina adaptors. Primers were removed using Agencourt AMPure XP bead cleanup (Beckman Coulter/Agencourt) and samples were amplified with 14 PCR cycles. Libraries were gel purified and quantified using a Qubit high sensitivity DNA kit (Invitrogen) and library quality was confirmed using Tapestation high sensitivity DNA tapes (Agilent Technologies). RNA Sequencing reactions were sequenced on an Illumina NextSeq sequencer (Illumina) according to the manufacturer's instructions, sequencing 50 bp reads. Analysis was performed using the CLC Genomics Workbench version 8.0.1 RNA-seq analysis software package implementing EdgeR for differential gene expression (Qiagen) or DESeq2. Reads were aligned (mismatch cost=2, insertion cost=3, deletion cost=3, length fraction=0.8, similarity fraction=0.8) to the mouse genome. Gene counts were loaded in R environment and differential gene expression was assessed with DESeq2. Gene-e (Broad Institute) was used to generate heatmaps. G:Profiler was used for pathway analysis using default settings.

## Single cell RNA sequencing

Draining lymph nodes from NP-OVA vaccinated Il21^FM/Rep mice (Tg(*Il21*^cre) *Rosa26*^Lox-STOP-Lox-TdTomato *Il21*^VFP) were harvested 9 days post-immunization, pooled in a single cell suspension and stained with fluorochrome-labeled cell surface antibodies. Live CD4⁺CD19⁻CXCR5⁺GITR⁻ cells were sorted based on VFP and TdTomato expression into Tfh-Prog (VFP⁻TdTomato⁻), Tfh-Trans (VFP⁺TdTomato⁻), Tfh-Full (VFP⁺TdTomato⁺) and Tfh-Ex (VFP⁻TdTomato⁺). Conventional T cells (CD4⁺CD19⁻CXCR5⁻GITR⁻) were also sorted as a control. Sorted cells were stained with distinct barcoded antibodies (Cell-Hashing antibody, TotalSeq-C, Biolegend) as previously described[41]. Next, cells from each condition were pooled together and resuspended in PBS 0.4% BSA at a concentration of 2000 cells/μl. Samples were subsequently loaded onto a single lane (Chromium chip K, 10X Genomics) followed by encapsulation in a lipid droplet (Single Cell 5′kit V2, 10X Genomics) at the Brigham and Women's Hospital Single Cell Genomics Core. cDNA and library generation were performed according to the manufacturer's protocol. The 5′ mRNA library was sequenced to an average of 50,000 reads per cell, whereas the V(D)J library and HTO (Cell Hashing antibodies) library were both sequenced to an average of 5000 reads per cell, all using Illumina Novaseq. Reads were processed with Cell Ranger, and quantification was performed using the STAR aligner against the Mm10 transcriptome. CellRanger output data were loaded into the R programming environment and analyzed with the Seurat package. Sample demultiplexing and doublet exclusion were performed with the HTODemux function, and only singlets were selected for further analysis. Additional quality-control filtering was performed, imposing as thresholds unique UMI counts ≥ 500, gene counts ≥ 250, log-transformed genes per UMI > 0.8 and mitochondrial RNA content <20%. Count data were subjected to normalization and variance stabilization using the SCTransform function (v.2), based on the 3000 most variable genes and by concomitantly regressing cell-cycle phase, mitochondrial, and ribosomal mapping percentages. Additional filtering based on identity was applied after comparison of each cells with the Immunologic Genome Project dataset (using the SingleR pipeline), to exclude contaminating cells (e.g. B, CD8⁺ T, and NK cells)[42]. Uniform manifold approximation and projection (UMAP) was used for dimensionality reduction according to the standard Seurat pipeline, but TCR-related genes were excluded from the list of variable features to avoid clustering based on clonotype. Differential Gene Expression between multiplexed samples was computed with the DESeq2 model implemented in the Seurat FindAllMarkers command, with a log(fold-change) threshold of 0.1 and by excluding ribosomal genes from the plots. Module scores were calculated with the AddModuleScore function, using as input a gene set comprising upregulated genes in Tfh-Full compared to Tfh-Prog cells, derived from our bulk RNAseq dataset. Average gene expression among the multiplexed samples was obtained with the AverageExpression command from the Seurat library and scaled with 0 and 1 as thresholds. TCR clonotype analysis was performed with the scRepertoire package[43]. Pseudotime trajectory analysis was carried out with Monocle3[44], while RNA velocity was calculated with the python packages scVelo and velocyto, according to standard pipelines[45].

## ATAC sequencing

Il21^FM/Rep mice (Tg(*Il21*^cre) *Rosa26*^Lox-STOP-Lox-TdTomato *Il21*^VFP) were vaccinated with NP-OVA 9 days prior. Tfh-Prog (VFP⁻TdTomato⁻), Tfh-Trans (VFP⁺TdTomato⁻), Tfh-Full (VFP⁺TdTomato⁺) and Tfh-Ex (VFP⁻TdTomato⁺) as well as T conventional cells were sorted, washed and lysed utilizing the ATAC Lysis Buffer (ATAC-Seq Kit, Active Motif). Tagmentation and library preparation were performed according to the manufacturer's instructions. Samples were pooled after indexing and sequenced on a llumina NextSeq sequencer (Illumina) according to the manufacturer's instructions, sequencing paired-end 50 bp reads. Data analysis was performed as previously described[46]. Briefly, quality trimming and primer removal from the raw fastq files were performed using Trimmomatic (v3.9) using the following parameters: LEADING:15 TRAILING:15 SLIDINGWINDOW:4:15 and MINLEN:36. The trimmed reads were aligned to Mm10 genome using Bowtie2 (v2.4.5) using a maximum insert size of 1000. PCR duplicates were marked using Picard (2.18.7). The concordance of each biological condition was assessed by the average Pearson correlation across all pairwise combinations. Peak-calling was performed for each biological condition using MACS (v2.2.7.1) on merged bam files with a *q*-value threshold of 0.001. Consensus peaks from all biological conditions were then merged to create a single peak universe of 56,060 regions. Cut sites were extracted from each biological replicate and the number of cuts within each peak region was quantified (BEDtools v2.30.0) to generate a raw counts matrix. DESeq2 (v3.15) was used to normalize the counts matrix and perform differential accessibility analysis between all of the relevant comparisons. For any given comparison, an FDR cutoff of 0.05 was used to determine the differential ChARs. Gene-to-peak associations were determined using the GREAT software package (v3.0.0). ATAC-seq tracks were visualized using Integrative Genomics Viewer (v2.13.2). Motif enrichment analysis was performed using HOMER (v3.0) with default settings.

## Imaging

Draining LN were harvested and immediately fixed in 4% PFA at 4 °C. After 4 h, tissues were washed with PBS, left overnight in a 30% sucrose solution and embedded in Tissue-Tek® O.C.T. Compound. Twenty micron-thick slices were blocked for one hour (10% donkey serum + 0.3% Triton) and stained with primary unlabeled antibodies against CD4 (rabbit anti-mouse, 1:100, clone EPR19514, Abcam) and IgD (goat anti-mouse, 1:500, polyclonal, Novus Biologicals) overnight at 4 °C.

After extensive washing, slides were stained with secondary antibodies (minimally cross-reactive donkey anti-goat AlexaFluor-594 and donkey anti-rabbit AlexaFluor-647, both 1:1000, Jackson Immunoresearch), T- and B-cell activation antigen, Pacific Blue (1:100, clone GL7) and AlexaFluor-488 GFP-Booster (1:500, ChromoTek) for one hour at room temperature. After mounting, images were acquired on a Leica Stellaris 8 confocal microscope with a 10x objective and processed with Fiji (version 2.9.0).

## Statistical analyses

Statistical analyses were performed using GraphPad Prism version 9.0 or R version 4.0.0. Student's two-tailed unpaired $t$ tests and Mann−Whitney $U$-tests, or one-way ANOVA and Kruskal−Wallist tests were used to compare between-group differences in normally and non-normally distributed data. Significance was inferred at the 5% of probability level. The number of mice per group, the number of replicates per experiment, summary statistics, and measures of dispersion are indicated in the legend of each figure.

## Reporting summary

Further information on research design is available in the Nature Portfolio Reporting Summary linked to this article.

## Data availability

Source data are provided with this paper for all figures. The sequencing data generated in this study have been deposited in the GEO database under GSE225724. All data are available from the corresponding author on reasonable request. Source data are provided with this paper.

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

## Acknowledgements

We would like to thank Dr. Uta Hoepken and Dr. Garnett Kelsoe for kindly providing mice and reagents, and the MicRoN (Microscopy Resources on the North Quad) Core, the Brigham and Women's Hospital Center for Cellular Profiling, the Harvard Medical School Biopolymer facility for their support and assistance in this work. We would also like to thank the Beth Israel Deaconess Medical Center Flow Core staff for help with flow cytometry. This work was supported by grants from the NIH (R01AI153124 to P.T.S., R01AI158413 to P.T.S., P01AI056299 to P.T.S. and R21AI158175 to P.T.S.) and by grant funding from Merck Sharp & Dohme LLC, a subsidiary of Merck & Co., Inc. Rahway, NJ, USA (to P.T.S.). Portions of the schematic illustrations in Figs. 2, 5–7, S1 and S4–5 in this work were created with BioRender.com.

## Author contributions

M.A.P., C.B.C., B.L.H., E.D.B., G.R., R.L.C., H.Z., P.C. and J.M.L. performed experiments. T.R.R. and R.A. provided technical help. M.A.P., C.B.C. and P.T.S. performed analyses. A.D. and D.R.S. provided additional bioinformatic analysis of chromatin accessibility. M.A.P., C.B.C. and P.T.S. conceptualized and designed the study and wrote the manuscript. P.T.S. supervised the study and acquired funding and resources.

## Competing interests

The authors declare no competing interests.
