## [Peer Review File · Nature Communications]

Stepwise differentiation of follicular helper T cells reveals distinct developmental and functional statesEditorial Note: This manuscript has been previously reviewed at another journal that is not operating a transparent peer review scheme. This document only contains reviewer comments and rebuttal letters for versions considered at *Nature Communications*.

REVIEWER COMMENTS

Reviewer #1 (Remarks to the Author):

I appreciate the efforts of the authors in generating bulk RNASeq data on the VFP+ Tfh cells from control and Foxp1-deleted cells. However, the presented data in Sup. Fig 5 still remains superficial and does not provide a clear mechanistic link for Foxp1 function. As the Foxp1 part of the manuscript is regarded by the authors as a major finding (despite previously published work on Foxp1, as discussed before) and also highlighted in the cartoon in Fig. 7F, more data mining and most importantly functional validation is required. The same applies to the newly added gene expression analyses of Bcl6, Cxcr5, Pdcd1, Icos, Maf and Ccr7 extracted from the bulk RNASeq data set in Fig. 7d. There are only 2 samples in the control group and their values (and the ones of the KO cells) vary extremely, with no statistical analysis actually being possible like this. Thus, the authors cannot claim that they "found no changes in TfhFull-associated genes" here.

Reviewer #2 (Remarks to the Author):

This study uses novel genetic tools to further dissect the biology of murine Tfh cells, specifically with regards to the timing of their IL-21 production (using fate-mapper and reporter strategies). Unfortunately, although a considerable body of data is presented, there are relatively few definitive insights in my view.

Like reviewer 3, I feel that the use of CXCR5 and ICOS (which are upregulated early after T cell activation) for the Tfh gate poses a problem: The addition of PD1 data adds a further layer of information but does not solve this issue. BCL6 staining is unfortunately not possible for technical reasons.

I agree with reviewer 3's interpretations with respect to proliferation and receipt of TCR signals (points 2 & 3). The correlations being reported are not overly surprising and the author's response – that cessation of IL-21 production is accompanied by additional transcriptional changes – is also unsurprising since TCR signalling would be expected to alter multiple pathways.

Additional points:

The authors suggest clear differentiation of Tfh-Full cells from Tfh-Prog cells in the absence of Tfr (Fig 2), however the % of IL-21+ cells here seems quite low.

Wouldn't the authors expect to see movement from Tfh-Prog to Tfh-Full in the pseudotime analysis?

Fig 6d,e. Results look quite variable and there are more repeats for the Tfh-Prog than other populations – this is not very convincing.

The image in Supplemental Fig 3A is used to show that Tfh-Prog are still present in and

around GC. However in Fig 5C not all Tfh-Full are deleted in this model so how do they know that these aren't just remaining Tfh-Prog?

Overall, there are too many uncertainties around data interpretation to be confident that robust conceptual advances are being made.

Reviewer #1 (Remarks to the Author):

I appreciate the efforts of the authors in generating bulk RNASeq data on the VFP+ Tfh cells from control and Foxp1-deleted cells. However, the presented data in Sup. Fig 5 still remains superficial and does not provide a clear mechanistic link for Foxp1 function. As the Foxp1 part of the manuscript is regarded by the authors as a major finding (despite previously published work on Foxp1, as discussed before) and also highlighted in the cartoon in Fig. 7F, more data mining and most importantly functional validation is required. The same applies to the newly added gene expression analyses of Bcl6, Cxcr5, Pdcd1, Icos, Maf and Ccr7 extracted from the bulk RNASeq data set in Fig. 7d. There are only 2 samples in the control group and their values (and the ones of the KO cells) vary extremely, with no statistical analysis actually being possible like this. Thus, the authors cannot claim that they "found no changes in TfhFull-associated genes" here.

We thank Reviewer 1 for reviewing the most recent manuscript revision and appreciate the helpful suggestions. We agree that the *in vitro* FoxP1 deletion TfhFull RNAseq dataset does have some variability that limits a clear picture of how FoxP1 is functioning at the molecular level. This variability is inherent to the system since each sample is a biological replicate (different mice) and changes between mice are likely compounded during the *in vitro* TfhFull differentiation assay. In addition, the number of viable cells that we are able to resort at the end of the assay is in the thousands, which adds to variability because it translates to lower input bulk RNAseq. We sequenced 4 biological replicates in control samples but two of them did not pass QC cutoffs and were not included in downstream processing/analysis. We appreciate the suggestion by Reviewer 1 to data mine the data set in more detail. However, due to the variability in the assay (and our results below) we thought this approach would not yield robust results.

Since the *in vitro* experiments have inherent variability associated with the complexity of the *in vitro* systems, we performed new *ex vivo* experiments in which we assessed the transcriptional program of TfhProg or TfhFull after FoxP1 deletion. The rationale for this experiment is that an earlier timepoint before culture may uncover FoxP1-dependent transcriptional changes in a less variable manner and may better maintain physiologically-relevant biology. We added a TfhProg group (which was not included in the *in vitro* assessment) since we predicted FoxP1 to have a more substantial role during this stage due to the higher expression levels. We vaccinated and administered tamoxifen to $UBC^{Ert2Cre}FoxP1^{floxed}IL21^{VFP}$ or control mice and sorted TfhProg and TfhFull cells which were processed for bulk RNASeq. Biological quadruplicates were included per group. We found many more DEGs between FoxP1-deleted and control mice in TfhProg cells (505 DEGs) compared to TfhFull cells (37 DEGs). These results suggest that FoxP1 has a more profound role during the TfhProg stage compared to the TfhFull stage and also validate that the TfhFull that develop in the context of FoxP1 deletion are transcriptionally similar to control TfhFull cells. Since Reviewer #1's concern was originally that FoxP1-deleted TfhFull may not be transcriptionally TfhFull we addressed this specific point in more detail. We performed gene set enrichment analysis (GSEA) on control or FoxP1-deleted TfhFull using a gene module for TfhFull (developed from RNAseq in Figure 1 and subsequently used in Figure 3F). We found no significant difference in the TfhFull gene module between control or FoxP1-deleted TfhFull, whereas control TfhFull were enriched in the TfhFull gene module compared to control TfhProg which were included as a positive control (Figure P1, below). These data suggest that the TfhFull that develop with FoxP1 deletion have a TfhFull program. These data have been added to the revised manuscript in Supplemental Figure 6b.

Figure P1. FoxP1-deleted TfhFull cells have similar enrichment of the TfhFull gene module compared to control TfhFull cells.

Reviewer #1 also requested that we assess more deeply how FoxP1 controls Tfh fate decisions utilizing FoxP1 ChIP-seq. Unfortunately, we cannot perform ChIP-Seq in our systems due to limited cell numbers. However, as suggested by Reviewer #1 we can use published datasets. We utilized an extensive FoxP1 ChIP-seq dataset in Treg and T conventional cells generated by the Rudensky Lab (Konopacki et al. *Nat. Immunol.* 2019). Of all DEGs found in our control versus FoxP1-deleted TfhProg or TfhFull cells, 99 had some evidence that they could be bound directly by FoxP1 (Figure P2, below). Of the 505 DEGs in TfhProg, 97 genes were putative FoxP1 target genes, 2 of which were also DEGs in TfhFull. Of the 97 genes that were differentially expressed in the TfhProg stage with FoxP1 deletion and were FoxP1 target genes, some have been implicated in controlling immune responses. We performed additional experiments to assess some genes identified as being different (by Gene-E find marker tool) at the protein level by flow cytometry. Although *Cd55* and *Gpr183* (encoding EBI2) seemed to be different in TfhProg and are FoxP1 target genes, we found only a trend of decreased CD55 and EBI2 at the protein level in FoxP1-deleted TfhProg cells and there was substantial variability. We also assessed Ly6a and CD124/IL4ra at the protein level, since these showed evidence of alteration in FoxP1-deleted Tfh cells at the gene expression level, although this did not reach statistical significance. Ly6a had slight increases in FoxP1-deleted TfhProg and TfhFull, but this did not reach statistical significance. However, IL4ra showed increases in both TfhProg as well as TfhFull. These genes are known to be involved in T cell activation programs, although their specific roles are not well understood. Some of these data have been added to Supplemental Figure 6.

Figure P2. FoxP1 deletion leads to more substantial changes during the TfhProg stage.

Beyond exploring FoxP1-target genes, we also assessed gene ontology pathways (using gProfiler) for DEGs in FoxP1-deleted versus control TfhProg cells. Interestingly, E2F-1 and FoxN4 transcription factor binding motifs were increased, which were also found in the bulkRNASeq dataset from *in vitro* Foxp1 deleted TfhFull cells (Figure P3, below). Moreover, E2F pathways were previously shown to be altered in CD8 T cells (Wei et al. J. Immunol. 2016) and Tregs (Ren et al. Plos Biol. 2019) in a FoxP1-dependent manner, although this has not been shown in Tfh cells. Interestingly, Bcl6B was also a hit in the analysis. Bcl6B binds to the same motifs as, and may form complexes with, Bcl6 in targeting DNA for regulation (Takenaga et al. *Biochem Biophys Res Commun* 2003). Moreover, Bcl6 and E2F have substantial overlap in genes they bind to (Ci et al. Blood 2009). Interestingly, CD55 and Il4ra mentioned above are known targets of E2F and/or Bcl6. Additionally, FoxN4 binding, which was also a hit in the motif analysis, is also a known Bcl6 regulated gene (GSE28737). Together, these data indicate that FoxP1 may regulate Bcl6 pathway genes to control Tfh stages.

source	term_name	term_id	adjusted_p_value	source	term_name	term_id	adjusted_p_value
TF	Factor: ZF5; motif: GSGCGCGR	TF:M00716	1.11E-23	TF	Factor: Elk-1; motif: NCCGGAAGTGN	TF:M10219	4.38E-12
TF	Factor: ZF5; motif: GSGCGCGR; match class: 1	TF:M00716_1	7.41E-22	TF	Factor: TEL1; motif: CNCGGAANN	TF:M01993	5.45E-12
TF	Factor: E2F; motif: GCGGSG	TF:M00803	1.03E-21	TF	Factor: XBP-1; motif: WNGMCACGTC	TF:M01770	5.68E-12
TF	Factor: FOXN4; motif: NNWANNCGWMC GCGTCNNNNMT	TF:M04662	7.51E-21	TF	Factor: Kaiso; motif: SARNYCTCGGAGAN; match class: 1	TF:M10276_1	1.06E-11
TF	Factor: E2F-1; motif: GNGGGCGGGRMN	TF:M10209	2.08E-20	TF	Factor: ERG; motif: ACCGGAART	TF:M01752	2.24E-11
TF	Factor: FOXN4; motif: NNWANNCGWMC GCGTCNNNNMT; match class: 1	TF:M04662_1	9.97E-20	TF	Factor: Elk-1; motif: NCCGGAAGTGN; match class: 1	TF:M10219_1	2.24E-11
TF	Factor: ZF5; motif: NRNGNGCGGCW/N; match class: 1	TF:M03333_1	1.01E-18	TF	Factor: E2F-4; motif: NTTTCSGCC	TF:M07380	2.45E-11
TF	Factor: Elk-1; motif: ACCGGAARTN; match class: 1	TF:M01981_1	4.05E-18	TF	Factor: ERG; motif: ACCGGAART; match class: 1	TF:M01752_1	3.22E-11
TF	Factor: PEA3; motif: RCCGGAAGYN; match class: 1	TF:M01991_1	8.84E-18	TF	Factor: c-Ets-1; motif: NNNRCCGGAWRYNNNN	TF:M01078	9.17E-11
TF	Factor: Kaiso; motif: SARNYCTCGGAGAN	TF:M10276	1.04E-17	TF	Factor: BEN; motif: CAGGGRNV; match class: 1	TF:M01240_1	1.03E-10
TF	Factor: GABP-alpha; motif: CTTCCK	TF:M01660	1.13E-17	TF	Factor: BCL6B; motif: NNNNCCGCCWNNNN	TF:M02844	1.22E-10
TF	Factor: ZF5; motif: NRNGNGCGGCW/N	TF:M03333	4.99E-17	TF	Factor: Ets2; motif: ACCGGAARTN	TF:M01989	1.31E-10
TF	Factor: ZF5; motif: GYCGCGCARNGC/N	TF:M02933	2.40E-15	TF	Factor: SAP-1a; motif: NRRCCGGAAGYRN	TF:M10220	2.14E-10
TF	Factor: E2F-1; motif: GNGGGCGGGRMN; match class: 1	TF:M10209_1	4.10E-15	TF	Factor: Ets2; motif: ACCGGAARTN; match class: 1	TF:M01989_1	2.42E-10
TF	Factor: Elk-1; motif: ACCGGAARTN	TF:M01981	6.79E-15	TF	Factor: E2F-4; motif: GCGGAAANA	TF:M02090	6.34E-10
TF	Factor: Elk-1; motif: NNNNCCGGAARTNN	TF:M00025	4.54E-14	TF	Factor: Sp1; motif: NGGGGGGGGGN	TF:M07395	6.55E-10
TF	Factor: PEA3; motif: RCCGGAAGYN	TF:M01991	1.72E-13	TF	Factor: GABP-alpha; motif: CTTCCK; match class: 1	TF:M01660_1	8.52E-10
TF	Factor: c-ets-1; motif: ACCGGAARTN	TF:M01986	2.51E-13	TF	Factor: BEN; motif: CAGGGRNV	TF:M01240	8.85E-10
TF	Factor: Fom1; motif: RGAMGC	TF:M12470	4.29E-13	TF	Factor: ELF4; motif: CCCGGAARTN	TF:M01979	1.15E-09
TF	Factor: E2F-3; motif: GCGGGGN	TF:M02089	1.02E-12	TF	Factor: E2F-1; motif: NKTSSCGC	TF:M00428	1.21E-09
TF	Factor: Elk-1; motif: NNNNCCGGAARTNN; match class: 1	TF:M00025_1	2.51E-12				

Figure P3. Pathway analysis of transcription factor motifs enriched in FoxP1-deleted TfhProg cells using gProfiler.

To assess Bcl6 pathways in more detail, we utilized the only Bcl6 ChIP-seq dataset available in mice from the Dong Lab (Liu et al. *Cell Rep.* 2016), even though it has some limitations because Bcl6 is notoriously difficult to study by ChIP-Seq (Choi et al. *Trends Immunol.* 2021). We annotated the DEGs from FoxP1-deleted TfhProg and TfhFull with Bcl6 bound genes and found 7 in TfhProg and 2 in TfhFull (Figure P4, below). Together these data highly suggest, but do not necessarily prove, that FoxP1 may control TfhProg by regulating Bcl6-associated pathways. These data have been added to the manuscript in Supplemental Figure 6. These results make sense considering the reciprocal nature of Bcl6 and FoxP1 expression in TfhProg and TfhFull, as shown in Figure 3h of the manuscript.

Figure P4. Overlap of Bcl6 bound genes and FoxP1 regulated genes.

Together these data implicate a reciprocal relationship between FoxP1 and Bcl6 and that Foxp1 may dampen Bcl6-associated pathways to insulate TfhProg. This hypothesis would only make sense if Bcl6 had an important function in stabilizing the TfhFull stage. Although Bcl6 is a known master regulator of Tfh cells that also controls CXCR5 expression, it is unclear if Bcl6 is required to specifically maintain the effector Tfh Program after development. Therefore, we performed new experiments in which we deleted Bcl6 *only* during the TfhFull stage. For this we immunized *Il21^{Cre}Bcl6^{flxed}Rosa26^{YFP}* or control mice and assessed TfhFull cells 12 days later. Bcl6 deletion during the TfhFull stage caused a drastic reduction in Tfh cells, with a profoundly disproportionate decrease in TfhFull cells (Figure P5, below). We also found a substantial decrease in the frequency of GC B cells. These data indicate that Bcl6 is essential to maintain TfhFull cells after they develop.

Figure P5. Bcl6 is required to stabilize the TfhFull stage.

Together, these new experiments help reveal mechanisms controlling Tfh developmental transitions. In summary, our new data suggests that FoxP1 acts predominantly during the TfhProg to stabilize this stage and that FoxP1-deleted TfhFull have few transcriptional changes. We hypothesize this is because Bcl6 dominates in the TfhFull stage and suppresses the effects of what little FoxP1 is left during this stage. We also found that Bcl6 is an essential stabilizing factor of the TfhFull stage. These data suggest that FoxP1 and Bcl6 reciprocally modulate each other's effects similarly to Blimp1 and Bcl6. However, it is important to point out that we do not have the tools to assess if FoxP1 *only* works through modulating Bcl6-associated pathways. Moreover, many of these effects could be indirect, such as through E2F transcription factor regulation. We have included additional discussion of these points, including the limitations of our data and interpretation, in the revised manuscript.

Reviewer #2 (Remarks to the Author):

This study uses novel genetic tools to further dissect the biology of murine Tfh cells, specifically with regards to the timing of their IL-21 production (using fate-mapper and reporter strategies). Unfortunately, although a considerable body of data is presented, there are relatively few definitive insights in my view.

Like reviewer 3, I feel that the use of CXCR5 and ICOS (which are upregulated early after T cell activation) for the Tfh gate poses a problem: The addition of PD1 data adds a further layer of information but does not solve this issue. BCL6 staining is unfortunately not possible for technical reasons.

We thank Reviewer 2 for reviewing the manuscript revision and for their comments. We agree that Bcl6 staining (and fate mapping) would be a better approach to identify/fate map Tfh developmental trajectories. However, as suggested by Reviewer 2, this strategy is impossible. We believe our approach, although imperfect, allows us to uncover previously unappreciated developmental stages in Tfh cells. We have included additional language in the results/discussion section on pages 5 and 22 of the revised manuscript about the limitations of our genetic approaches in uncovering Tfh developmental stages.

I agree with reviewer 3's interpretations with respect to proliferation and receipt of TCR signals (points 2 & 3). The correlations being reported are not overly surprising and the author's response – that cessation of IL-21 production is accompanied by additional transcriptional changes – is also unsurprising since TCR signalling would be expected to alter multiple pathways.

We thank Reviewer 2 for this comment. We agree that loss of TCR signals may explain the formation of TfhEx cells from TfhFull and that this interpretation has strong rationale. Although formation of TfhEx may not be surprising, we believe characterizing this population is important as it has implications for how the germinal center is maintained. We have added additional discussion about this point on page 22-23 of the revised manuscript.

Additional points:

The authors suggest clear differentiation of Tfh-Full cells from Tfh-Prog cells in the absence of Tfr (Fig 2), however the % of IL-21+ cells here seems quite low.

We thank Reviewer 2 for this observation. We developed the TfhProg to TfhFull in vitro differentiation assay to assess mechanisms controlling this developmental stage transition. Although the frequency of cells that commit to the TfhFull stage is relatively small shown in Fig. 4g-I (~4%) this is still important as it allows us to assess synchronized TfhFull development which is impossible to

accomplish *in vivo*. It is possible that our *in vitro* system does not recapitulate *all* cell types/signals as in an *in vivo* GC reaction which may slow/limit TfhFull development. However, this makes sense as our data shows that Tfh developmental trajectories require more than just TCR signals. It is also important to point out that ~4% TfhFull development from TfhProg is also consistent with findings in Figure 7 in control mice. The data in Figure 7 also shows that Foxp1 signaling is one factor inhibiting the TfhProg to TfhFull transition *in vitro*.

Wouldn't the authors expect to see movement from Tfh-Prog to Tfh-Full in the pseudotime analysis?

We thank Reviewer 2 for this comment. Yes, we do expect movement from TfhProg to TfhFull in pseudotime analysis, and that is what we observe. As shown below there is movement towards TfhFull cells, predominantly clusters 2 and 6 (Figure P6, below). To illustrate this in more detail we also generated a box plot for pseudotime for each condition which shows TfhFull and TfhEx as having the largest pseudotime and TfhProg having the least. We have added the boxplot to the revised manuscript in Supplemental Figure 2.

Figure P6. Pseudotime analysis of Tfh developmental stages from scRNASeq data.

Fig 6d,e. Results look quite variable and there are more repeats for the Tfh-Prog then other populations – this is not very convincing.

We thank Reviewer 2 for bringing this to our attention and acknowledge high variability in these assays. The transfer experiment in Figure 6 is a technically challenging experiment that requires two adoptive transfers and cell sorting of individual stage cells to assess expansion and functionality of TfhProg. These assays require mice with a rare combination of alleles that is very difficult to breed (OT-II⁺II21^{Cre}Rosa26^{TdTomato}II21^{VFP}) limiting the number of experiments we can perform. We believe the variability in these assays is due to differential engraftment after adoptive transfer of TfhProg, most likely due to variability associated with cell sorting. However, it is also possible that there is a differential amount of expansion in the primary recipient which alters ability of TfhProg to expand in secondary recipient. The Tfh stage cells in these experiments are sorted from ~12 mice. We have more replicates for the TfhProg because cell numbers are normalized for the transfer and we can sort more TfhProg than other cell subsets due to their total numbers at this timepoint. This timepoint was chosen intentionally because the main purpose of the figure is to explore expansion potential of TfhProg. In the revised manuscript we have included details about variability and indicate that this assay is designed to assess behavior in TfhProg and may be limited when assessing other cell subsets due to variability on page 18 of the revised manuscript.

The image in Supplemental Fig 3A is used to show that Tfh-Prog are still present in and around GC.

However in Fig 5C not all Tfh-Full are deleted in this model so how do they know that these aren't just remaining Tfh-Prog?

We thank Reviewer 2 for this comment. We know that the remaining TfhFull in Figure 5C are not TfhProg because they do not express the YFP fate-mapping allele. TfhProg are defined based on lack of YFP expression indicating that these cells have never expressed IL-21. Whereas the remaining TfhFull in Fig. 5C express the YFP fate mapping allele.

Overall, there are too many uncertainties around data interpretation to be confident that robust conceptual advances are being made.

We appreciate this point of view from Reviewer 2. Although our systems may not be perfect, we believe that uncovering Tfh developmental stages in this way is worth-while and advances the field. Throughout the revised manuscript we have highlighted the uncertainties around data interpretation and have toned down claims so that the reader can understand the technical limitations of the study.

REVIEWERS' COMMENTS

Reviewer #1 (Remarks to the Author):

As previously pointed out, the bulk RNA sequencing data with n=2 in the control condition exhibits great variability that is too high to make any conclusions from the data and should probably not be published in its current form.

The ex vivo data adds more transcriptional analyses to the study. Suppl. Fig 6d should be added to the main figure. The Chip-seq re-analysis adds additional descriptive data to Suppl. Fig. 6C/E. Unlike in the rebuttal letter, there is no description and discussion of the newly added flow cytometry data on Ly6a (unchanged) and CD124 (differently expressed between Prog Ctrl and FL) (Fig. S6d) in the manuscript.

Concerning the newly added data in Fig. 7h: The requirements of Bcl6 for Tfh-full maintenance have actually been previously shown, for example for CXCR5+ memory Tfh cell maintenance (PMID: 25071203), preTfh/GC Tfh cell maintenance (PMID: 33027650), and memory/Tfh cells (PMID: 34792530) using tamoxifen induced Bcl6 deletion in CD4 T cells or by Ox40 driven Cre. Thus, the new data presented in the revised version of the manuscript concerning Tfh-full are not as new as claimed by the authors, yet complement those previous studies by using a different Cre driver (IL-21 CreERT2). The description of their findings "Although Bcl6 is thought to be the master transcription factor for Tfh cells, the role of Bcl6 after initial differentiation and during specific Tfh stages has not been studied in detail. (...) Together these data demonstrate the essential role of Bcl6 in maintaining the Tfh-Full stage (...) Nevertheless, the profound loss of Tfh-Full with Bcl6 deletion suggest that Bcl6 is essential to maintain Tfh-Full cells." should be updated in light of the current literature.

Reviewer #2 (Remarks to the Author):

This is not a particularly convincing rebuttal. Regarding my statement "Like reviewer 3, I feel that the use of CXCR5 and ICOS (which are upregulated early after T cell activation) for the Tfh gate poses a problem" there has not been a meaningful effort to acknowledge this limitation in the manuscript. The text inserted on p.5 is not in a relevant place (and does not discuss limitations of CXCR5/ICOS staining). The logical place to insert such an acknowledgement would be after the sentence "Tfh cells were defined as CD4+CD19-GITR-ICOS+CXCR5+ and then subdivided into YFPexpressing and non-expressing cells (Fig. 1a)."

Having reviewed the current version, my view that the manuscript offers lots of data yet few definitive insights has not really changed. The authors have explained the technical reasons for the limitations and reinforced their opinion that the manuscript offers important conceptual advances. I am happy for the community to judge this for themselves and draw their own conclusions.

REVIEWERS' COMMENTS

Reviewer #1 (Remarks to the Author):

As previously pointed out, the bulk RNA sequencing data with n=2 in the control condition exhibits great variability that is too high to make any conclusions from the data and should probably not be published in its current form.

We thank Reviewer #1 for this suggestion. To address this we have moved the in vitro bulkRNASeq on FoxP1-deleted TfhFull cells to Revised manuscript supplemental figure 5 so that it is not in the main figures. We opted for including the data in the supplement instead of omitting it completely to ensure full transparency of data that was generated during the manuscript review process, to indicate different strategies used to assess FoxP1 regulation, and so that the field is aware the data is available and may be of potential use by other investigators studying FoxP1. However, we fully acknowledge that the data is highly variable due to the low number of replicates. Therefore, we have added additional language of the high variability and low n number of the study in the revised manuscript text on page 20.

The ex vivo data adds more transcriptional analyses to the study. Suppl. Fig 6d should be added to the main figure. The Chip-seq re-analysis adds additional descriptive data to Suppl. Fig. 6C/E. Unlike in the rebuttal letter, there is no description and discussion of the newly added flow cytometry data on Ly6a (unchanged) and CD124 (differently expressed between Prog Ctrl and FL) (Fig. S6d) in the manuscript.

We thank Reviewer #1 for this suggestion. As recommended, we have added the data from Fig. S6d into the main revised manuscript figure 7e. We have also added a description of the data into the revised manuscript file on page 21.

Concerning the newly added data in Fig. 7h: The requirements of Bcl6 for Tfh-full maintenance have actually been previously shown, for example for CXCR5+ memory Tfh cell maintenance (PMID: 25071203), preTfh/GC Tfh cell maintenance (PMID: 33027650), and memory/Tfh cells (PMID: 34792530) using tamoxifen induced Bcl6 deletion in CD4 T cells or by Ox40 driven Cre. Thus, the new data presented in the revised version of the manuscript concerning Tfh-full are not as new as claimed by the authors, yet complement those previous studies by using a different Cre driver (IL-21 CreERT2). The description of their findings “Although Bcl6 is thought to be the master transcription factor for Tfh cells, the role of Bcl6 after initial differentiation and during specific Tfh stages has not been studied in detail. (...) Together these data demonstrate the essential role of Bcl6 in maintaining the Tfh-Full stage (...) Nevertheless, the profound loss of Tfh-Full with Bcl6 deletion suggest that Bcl6 is essential to maintain Tfh-Full cells.” should be updated in light of the current literature.

We thank Reviewer #1 for this comment. We agree that the requirement of Bcl6 for the maintenance of Tfh cells has been previously shown using tamoxifen/inducible deletion on all CD4 or Tfh cells. We apologize for omitting these references. However, the rationale for our experiment was meant to indicate that Bcl6 has not been deleted in specific subsets of Tfh cells, in particular Tfh-Full cells, previously. In this way we were able to clarify the role of Bcl6 on Tfh-Full cells without deleting in Tfh-Prog cells. In the revised manuscript we have added references to the previous work on Bcl6 in Tfh maintenance and have also toned down our claims as suggested by Reviewer #1 in both the revised manuscript main text (on page 22) and discussion (on page 24).

Reviewer #2 (Remarks to the Author):

This is not a particularly convincing rebuttal. Regarding my statement “Like reviewer 3, I feel that the use of CXCR5 and ICOS (which are upregulated early after T cell activation) for the Tfh gate poses a problem” there has not been a meaningful effort to acknowledge this limitation in the manuscript. The text inserted on p.5 is not in a relevant place (and does not discuss limitations of CXCR5/ICOS staining). The logical place to insert

such an acknowledgement would be after the sentence “Tfh cells were defined as CD4+CD19-GITR-ICOS+CXCR5+ and then subdivided into YFPexpressing and non-expressing cells (Fig. 1a).”

We thank Reviewer #2 for this suggestion and apologize for not suitably addressing this concern during the last revision. We have inserted a clear description that CXCR5 and ICOS expression for Tfh gating in Figure 1 has limitations after the specific sentence mentioned by Reviewer #2 in the revised manuscript which can be found on page 5.

Having reviewed the current version, my view that the manuscript offers lots of data yet few definitive insights has not really changed. The authors have explained the technical reasons for the limitations and reinforced their opinion that the manuscript offers important conceptual advances. I am happy for the community to judge this for themselves and draw their own conclusions. understand the technical limitations of the study.

We thank Reviewer #2 for their time in reviewing the manuscript. We appreciate the feedback from Reviewer #2 and their decision to let the field judge the impact of the findings for themselves.